# Universal and taxon-specific trends in protein sequences as a function of age

**Jennifer E James[1]\*, Sara M Willis[1], Paul G Nelson[1], Catherine Weibel[2,3], Luke J Kosinski[4], Joanna Masel[1]\***

[1]Department of Ecology and Evolutionary Biology, University of Arizona, Tucson, United States; [2]Department of Physics, University of Arizona, Tucson, United States; [3]Department of Mathematics, University of Arizona, Tucson, United States; [4]Department of Molecular and Cellular Biology, University of Arizona, Tucson, United States

**Abstract** Extant protein-coding sequences span a huge range of ages, from those that emerged only recently to those present in the last universal common ancestor. Because evolution has had less time to act on young sequences, there might be 'phylostratigraphy' trends in any properties that evolve slowly with age. A long-term reduction in hydrophobicity and hydrophobic clustering was found in previous, taxonomically restricted studies. Here we perform integrated phylostratigraphy across 435 fully sequenced species, using sensitive HMM methods to detect protein domain homology. We find that the reduction in hydrophobic clustering is universal across lineages. However, only young animal domains have a tendency to have higher structural disorder. Among ancient domains, trends in amino acid composition reflect the order of recruitment into the genetic code, suggesting that the composition of the contemporary descendants of ancient sequences reflects amino acid availability during the earliest stages of life, when these sequences first emerged.

\*For correspondence:
jejames@email.arizona.edu (JEJ);
masel@email.arizona.edu (JM)

**Competing interests:** The authors declare that no competing interests exist.

## Introduction

New protein-coding genes emerge over time, either through a combination of rearrangement, duplication, and divergence from ancestrally coding sequences, or de novo from previously non-coding DNA (for a review see *Van Oss and Carvunis, 2019*, *Long et al., 2003*). Although genes clearly originated de novo during the emergence of life, de novo gene birth after this unique event was once believed to be extremely rare (*Jacob, 1977*; *Keese and Gibbs, 1992*; *Zuckerkandl, 1975*). Today, de novo gene birth is increasingly acknowledged to be real and important, with well-documented examples confirmed in diverse taxa (*Baalsrud et al., 2018*; *Cai et al., 2008*; *Khalturin et al., 2009*; *Milde et al., 2009*; *Zile et al., 2020*), despite the technical challenges associated with correctly identifying de novo genes (*McLysaght and Hurst, 2016*). Compared to genes born early in the history of life, recently emerged de novo genes have had little time to evolve and adapt. If their properties converge only slowly to those of ancient genes, this creates trends in gene properties with age (*Neme and Tautz, 2013*).

It has been claimed that younger genes encode shorter and faster evolving proteins (*Lipman et al., 2002*; *Toll-Riera et al., 2012*) that are more disordered, containing fewer hydrophobic amino acids (*Wilson et al., 2017*), which are more clustered together along the primary sequence (*Foy et al., 2019*). *Foy et al., 2019* suggest that the two latter trends, in hydrophobicity and in its clustering, together indicate that evolution is slow to find the sophisticated folding strategies employed by older proteins, which minimize their propensity to form aggregates within cells despite relatively high overall hydrophobicity.

The universality of some of these trends has, however, been questioned. In particular, some authors have found that in some taxa, such as *Saccharomyces cerevisiae* (*Carvunis et al., 2012*; *Vakirlis et al., 2018*), young genes are in fact less disordered than older genes. Random sequences with high %GC tend to encode polypeptides with higher intrinsic structural disorder (ISD) (*Ángyán et al., 2012*). Differences in genomic GC content between species might therefore shape disorder at the time of gene birth differently (*Basile et al., 2016*; *Van Oss and Carvunis, 2019*). Because GC content evolves over relatively short timescales, GC content could explain differences in de novo genes among species, but less so among broad taxonomic groups.

Alternatively, the low disorder of young 'protogenes' may be an artifact of including sequences that do not meet the evolutionary definition of functionality, which requires that the loss of the gene be deleterious, that is, have a selection coefficient of less than zero (*Graur et al., 2013*). In a species whose %GC gives rise to low structural disorder in non-functional protogenes, the pooling of high-disorder functional and low-disorder non-functional genes in varying proportions can create a spurious trend with age. Indeed, the direction of the disorder trend in *S. cerevisiae* was reversed when dubious gene candidates were removed from the data of *Carvunis et al., 2012* (*Wilson et al., 2017*).

Nevertheless, with phylostratigraphy so far studying a limited set of relatively well-annotated focal species, it remains possible that trends in protein properties as a function of age are not universal, and instead depend on taxonomic group. The forces that shape gene birth could differ across lineages because the first imperative to 'do no harm' operates differently as a function of differences in proteostasis machinery. Alternatively, different amino acids might be abundant, and hence cheap to use, in different taxa. We therefore set out to test whether trends in protein properties are the same vs. different across sequences of different ages and across different species, and which forces might underlie any differences.

Identifying phylostratigraphy trends as a function of gene age requires age estimation. Gene ages are based on the date of the most basal node in the phylogeny of lineages containing homologs (*Domazet-Loso et al., 2007*). But when sequences are highly divergent, programs used to detect homologs such as BLASTp (*Altschul et al., 1990*) are prone to false negatives, and thus underestimate gene age (*Elhaik et al., 2006*; *Jain et al., 2019*; *McLysaght and Hurst, 2016*; *Moyers and Zhang, 2015*; *Wolfe, 2004*). This is particularly problematic when studying protein properties such as length, evolutionary rate, and degree of conserved structure, because these properties themselves directly impact our ability to detect sequence similarity. Simulation studies have shown that these biases in homology detection have the potential to drive spurious phylostratigraphic trends (*Elhaik et al., 2006*; *Moyers and Zhang, 2015*; *Moyers and Zhang, 2016*). However, the number of 'true' young genes identified by phylostratigraphy studies may dwarf the numbers of genes that are falsely identified as young due to errors in homology detection, making that bias insufficient to explain previously observed trends (*Domazet-Lošo et al., 2017*). Statistically correcting for length and evolutionary rate did not remove observed trends in ISD (*Wilson et al., 2017*) or clustering (*Foy et al., 2019*).

Simulation studies suggest that reducing false negatives will reduce the strength of trends driven by homology detection bias (*Moyers and Zhang, 2018*). Therefore, if we improve the sensitivity of homology detection methods, this is predicted to result in weaker trends in protein properties with age if previously reported trends in protein properties, such as disorder and clustering, were spurious. In contrast, if previously reported trends are not artifacts, we predict that with increased sensitivity they will get stronger. This is because many false negatives may be random errors, which reduce power in detecting trends (*McLysaght and Hurst, 2016*), rather than systematic errors, which can create spurious trends.

Whatever the issues with homology detection, abandoning it is not a viable option. Because homologs share an evolutionary history, they are not statistically independent. But many –omic studies treat genes as independent datapoints and are thus flawed due to pseudoreplication. Phylostratigraphy can solve this problem by making each datapoint represent the independent evolutionary origin of a protein sequence, avoiding phylogenetic confounding (*Felsenstein, 1985*; *Thornton and DeSalle, 2000*). It is therefore critical that we improve upon the BLAST-based homology detection methods used within phylostratigraphy, rather than abandon the effort.

Fortunately, more recently developed methods, such as PSI-BLAST (*Altschul and Koonin, 1998*) and HMMER3 (*Finn et al., 2011*), are more sensitive to distant homology, while still being

computationally efficient. Both are multiple sequence methods, which have long been known to be more sensitive to distant homologs than pairwise methods (*Park et al., 1998*). PSI-BLAST is a heuristic method that constructs a position-specific score matrix from the alignment of known homologs, which can then be used to iteratively search a protein database for additional homologs. HMMER instead constructs a profile HMM – a probabilistic model of the alignment – and is thus based on a formal statistical framework, yielding superior treatment of indels (*Eddy, 2011*). Profile HMM based methods perform significantly better than sequence based methods in comparative studies on diverse types of sequence data, with little difference in the rate of false positives (*Freyhult et al., 2010*; *Madera and Gough, 2002*).

Unfortunately, because both methods are based on iterative searches, both are vulnerable to model corruption, where a single false positive hit to a non-homologous sequence will pull in many more non-homologous sequences, potentially snowballing over multiple iterations (*Pearson et al., 2017*). Automated pipelines for whole-genome analysis are technically challenging, and HMMER is generally used with manual supervision of alignments.

Here we increase the sensitivity of homology detection during phylostratigraphy by using the manually curated pfam database, which was constructed using a HMMER3 pipeline. For each pfam, an HMM was built from high quality seed alignments, homologs were then found, and matches above a curated threshold were realigned back to the HMM (*El-Gebali et al., 2019*).

Pfams are intended to correspond to protein domains, which are structural units, capable of folding independently (*Holm and Sander, 1994*; for a review discussing domain definitions and identification, see *Ponting and Russell, 2002*). These domains are often considered to be the 'true' units of homology, with full proteins made up of one or more domains in a variety of different combinations (*Bagowski et al., 2010*; *Chothia et al., 2003*; *Koonin et al., 2000*; *Moore et al., 2008*). The pfam database has developed over time and now curates sets of clearly homologous sequences that are no longer required to correspond to compact folds. We continue, however, to use the term 'domain' in conjunction with pfams.

Here we apply HMMER3-based phylostratigraphy to a broad range of taxa. A key methodological innovation is to use pfam domains rather than BLASTp as our unit of homology. A second innovation is that instead of studying only sequences that are present in one or a few focal species, and using other species only to assign age of origin, we make full use of 435 fully sequenced species, effectively treating most of them as focal species. This not only increases our power, but also lets us evaluate how general trends are across different phylogenetic groups. We use only those pfams that are present in at least two species, a quality filter that ensures the exclusion of non-genic contaminants from our data set. We also analyze patterns of protein evolution in full genes, dating these by the oldest pfams that they contain, making results comparable to earlier phylostratigraphies based on the oldest short segment BLASTp hits (*Albà and Castresana, 2005*; *Domazet-Loso et al., 2007*). Additionally, to avoid spurious trends most likely to be due to homology detection bias, we avoid the protein metrics that most strongly affect homology detection (i.e. evolutionary rate and length), and focus on trends in properties expected a priori to have less effect on homology detection, such as the hydrophobicity and the degree of clustering of hydrophobic amino acid residues.

Our large data set gives us unprecedented power to answer two key questions. First, which trends in long-term evolution are supported rather than being due to artifacts? Here our increased power enables us to detect finer-scale trends, such as changes in the frequencies of each of the 20 amino acids, rather than simply summary statistics such as ISD. Trends in ISD and clustering get stronger with improved methods, consistent with being biological rather than due to artifacts. Second, are trends in long-term evolution universal across our tree, or specific to a particular group? To answer this, we compare trends among pfams that were born in animals, trends among pfams that were born in plants, and trends among pfams that were born prior to the origin of eukaryotes. We find that while trends in clustering are general across our data set, ISD and amino acid composition trends are specific to particular lineages – the latter compatible with specific biology rather than general artifacts of homology detection.

## Results

In this study, we analyzed all protein-coding genes containing at least one pfam annotated in the genomes of 435 species, with a total of 8209 pfams identified (see Materials and methods). We used

a variety of quality filters to exclude pfams that may be due to contaminants, annotation errors or horizontal gene transfer (see Materials and methods). Eukaryotic species were included in our data set if they were marked 'Complete' by GOLD (*Mukherjee et al., 2019*), and also present in TimeTree (*Hedges et al., 2006*), a phylogenetic database that we used to assign evolutionary ages to pfams. A phylogenetic tree summarizing the species used in this analysis is given in *Figure 1—figure supplement 1*.

Where not otherwise indicated, throughout our main results we focus on non-transmembrane pfams. It is a priori likely that transmembrane pfams experience different selective pressures as they evolve, given the very different biophysical characteristics of a lipid bilayer vs. the cytosol. Pooling is therefore not appropriate unless similarities have first been established.

To avoid pseudoreplication caused by phylogenetic confounding, we take the average of each protein property across all homologs of a pfam, and then treat each homologous set as a single data point in subsequent analysis (see Materials and methods). Phylostratigraphy assigns each such set of homologous pfams to an age class, dated using timetree. We then use linear regression to obtain slope, which provides our estimate of the effect size of the relationship between sequence properties and age. In order to compare our results with those of previous studies, we also conducted our analyses on full genes.

## Trends in ISD

Young genes (*Willis and Masel, 2018*; *Wilson et al., 2017*; *Foy et al., 2019*; *Mukherjee et al., 2015*) and domains (*Bornberg-Bauer and Albà, 2013*; *Buljan and Bateman, 2009*; *Ekman and Elofsson, 2010*; *Moore and Bornberg-Bauer, 2012*) have been reported to have high ISD, although some have claimed this depends on taxon (*Vakirlis et al., 2018*). We use IUPred to estimate ISD from amino acid sequence alone, allowing estimation across large-scale genomic data sets that contain many sequences of undetermined structure (*Mészáros et al., 2018*). We confirm that across our entire data set, ISD is higher in young pfams (*Figure 1*, linear model: $R^2 = 0.13$, $p = 3 \times 10^{-245}$) and genes (*Figure 1—figure supplement 2A*, linear model for all genes: $R^2 = 0.057$, $p = 1 \times 10^{-229}$).

Our improved methodology increased the steepness (i.e. effect size) of the relationship between gene ISD and gene age in mouse genes above that previously estimated by *Foy et al., 2019* (*Figure 1—figure supplement 2B*), from −0.028 to −0.050; note that mouse genes have a steeper slope than our results across all taxa. However, we note that age explains a relatively small proportion of the variance in ISD, with the remainder presumably driven by a combination of function, random biological variation, and measurement error (IUPred being an imperfect proxy for ISD or whatever other correlated biophysical property truly underlies the trend).

The slope of the relationship between ISD and age is 1.8-fold steeper for pfams (*Figure 1A*) than for full genes (*Figure 1—figure supplement 2A*); this is unsurprising, because whole genes are made up of combinations of sequences of different ages. The fact that protein domains are a more fundamental evolutionary unit than genes (*Bornberg-Bauer et al., 2005*; *Moore et al., 2008*), which is reflected in stronger relationships with pfam age, demonstrates an advantage of using a pfam-based phylostratigraphy approach. While this study focuses on pfam domains, we note that results are similar, but with smaller effect sizes, when calculated over genes.

To investigate how the ISD trend varies over time, we recalculated phylostratigraphy slopes over age-restricted subsets of our data. We also compared animal vs. plant lineages, given that these are the two kingdoms for which we have the most data (343 animal genomes and 87 plant genomes). The trend in ISD is not consistent (*Figure 1B*, *Figure 1—figure supplement 3*), but is instead driven by recent animal pfams (in which we include all pfams that emerged after the divergence of animals/fungi from plants, 1496 MYA, that are now present in animals).

There is no significant change in ISD over 'ancient' pfams (those that emerged prior to the last eukaryotic common ancestor [LECA], which is estimated to have existed around 2101 MYA). This does not mean that ISD is static; for example, the same domain in more exquisitely adapted vertebrates has higher ISD (*Weibel, 2020*).

There is also no change in ISD with age over recent plant pfams (i.e. all pfams found in plants that are younger than 1496 MY old). Note that we have relatively few plant-specific pfams in our data set compared to animal-specific pfams (*Figure 1—figure supplement 3*). This is likely due to differences in genome quality and annotation between the groups, and the corresponding lack of available plant genomes that meet our quality standards. However, it is also possible that these numbers reflect a

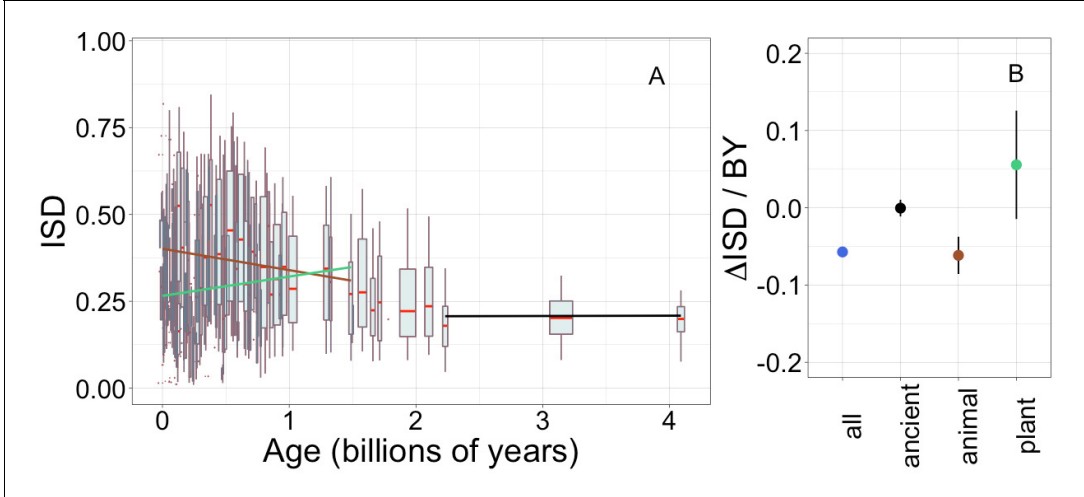

**Figure 1.** Young domains have high intrinsic structural disorder (ISD), but this trend is driven exclusively by recent animal domains. Results are for non-transmembrane pfams. (A) The brown linear regression was calculated for recent animal pfams (slope = −0.062, $R^2$ = 0.0097, p = 6 × 10$^{-7}$, n (number of pfams) = 2456), green for recent plant pfams (slope = 0.056, p = 0.1, n = 183), and black line over ancient pfams in all lineages (slope = −4.2 × 10$^{-4}$, p = 0.9, n = 3102). Data underlying animal and plant curves is visualized separately in *Figure 1—figure supplement 3*. Slopes represent the decrease in average IUPred2 score, that is, the predicted propensity of the average amino acid to be disordered, per billion years. Each data point consists of the average across all instances of homologous pfams, across all species in which it occurs. Phylostratigraphy assigns these to age classes, dated using timetree. To help visualize the data, every age class is represented in the figure by a weighted box plot, where the width of the plot indicates the number of pfams in that age class. The median is shown in red, with the boxes representing upper and lower quartiles (the 75th and 25th percentile), and the whiskers indicating 9 and 91 quantiles. For age classes with only a single pfam, values are presented as small red dots. For clarity of presentation our plots do not show outliers, although we note that these are included in our linear regression models. (B) Phylostratigraphy slopes for pfams calculated over different subsets of the data are plotted with their 95% confidence intervals. The point colors correspond to the regression slopes in (A).

The online version of this article includes the following figure supplement(s) for figure 1:

**Figure supplement 1.** Phylogenetic tree of all species used in this analysis.

**Figure supplement 2.** Intrinsic structural disorder (ISD) depends on age for whole genes, as previously reported by *Foy et al., 2019*.

**Figure supplement 3.** Young animal domains have high intrinsic structural disorder (ISD), while we have limited power to detect trends in young plant domains.

**Figure supplement 4.** Recalculating intrinsic structural disorder (ISD) after excising cysteine residues has very little effect on our results.

biological reality that animal proteomes are richer in domains (*Ramírez-Sánchez et al., 2016*). We also note that animal- and plant-specific pfams have different age distributions due to differences in the respective phylogenies, and thus the set of possible divergence times in these groups.

Recent animal pfams have higher mean disorder (0.36) than recent plant pfams (0.26) (Welch's t-test p = 4 × 10$^{-5}$, figure 1a shows how this result depends on age within the recent pfam categories), with the latter still higher than ancient pfams (0.21). The difference between animals and plants does not reflect differences in the birth process alone; even in ancient pfams that are shared by plants and animals, mean ISD is 0.27 in animals vs. 0.24 in plants, a smaller but still significant difference (Wilcoxon signed rank test [a paired, non-parametric test] on difference p = 0.005). These results are compatible with animal domains experiencing more selection for high ISD than plants; plant domains have consistently less disorder with the difference being more pronounced in young, lineage-specific domains. This may be because plants produce aggregation inhibiting molecules (*Velander et al., 2017*), thus reducing the risk that protein aggregation poses to plant cells (see Discussion).

## Trends in amino acid frequencies

IUPred scoring of ISD primarily reflects amino acid composition, with hydrophilicity being a major determinant of disorder (*Dosztányi et al., 2005b*; *Wilson et al., 2017*). Our larger data set has sufficient power to investigate age trends in the frequencies of each of the 20 amino acids individually,

rather than just trends in the single IUPred summary statistic. This can reveal which amino acids drive our ISD results, as well as other potentially interesting patterns in amino acid occurrence.

Trends in amino acid frequencies with age among ancient pfams are essentially identical whether they are assessed using only plant data or only animal data (*Figure 2A*, Spearman's ρ = 0.94, p = $6 \times 10^{-6}$, unweighted Pearson's R = 0.98, p = $8 \times 10^{-14}$). In subsequent analyses beyond *Figure 2*, we therefore pool data across all species in our data set to calculate the phylostratigraphy slopes among ancient pfams with higher resolution.

In contrast, recent animal pfams and recent plant pfams show different trends in amino acid frequencies with age, with slopes of similar magnitude (Welch's t-test on absolute slope values p = 0.3), but no correlation in value (*Figure 2B*, p = 0.8). Recent frequency trends are mostly unrelated to ancient trends, with no relationship for animals (*Figure 2C*, p = 0.8), and a weak correlation for plants that is not significant (*Figure 2D*, p = 0.2) and is entirely driven by cysteine.

In plants, cysteine has the steepest phylostratigraphy slope. *Miseta and Csutora, 2000* previously reported that %cysteine content decreased with 'complexity', from mammals, to plants, to a set of single celled organisms that included both photosynthesizing and non-photosynthesizing bacteria and archaea. Over all pfams, our results agree with the findings of Miseta and Csutora, in that cysteine is slightly more common in animals (2.5%) than in plants (2.2%). But surprisingly, young, plant-

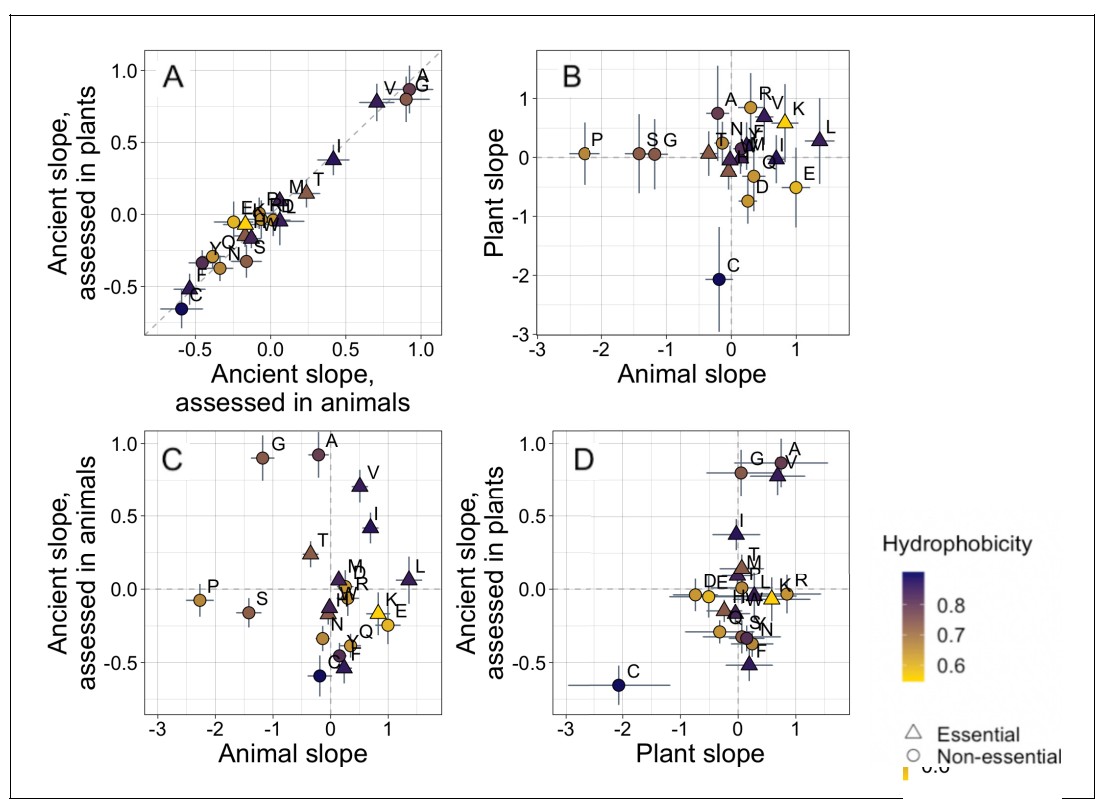

**Figure 2.** Trends in amino acid frequencies as a function of age differ across lineages. Results are shown for non-transmembrane pfams. Phylostratigraphy slopes are in units of the change in percentage points of an amino acid per billion years. 'Ancient' refers to pfams older than 2101 MY, assessed in only plant or only animal instances, whereas 'plant' and 'animal' slopes are calculated over pfams appearing after the divergence between the animal/fungi and plant lineages, 1496 MY, again assessed in only plant or only animal instances. Lines indicate the standard errors on the slopes. Points are color-coded by their hydrophobicity, as measured by 1-mean relative solvent accessibility (RSA) (*Tien et al., 2013*), such that buried, hydrophobic amino acids are dark purple, and exposed, hydrophilic amino acids are yellow. RSA scores are based on the water-accessible surface area of amino acids in a training set of proteins of known structure. Amino acid shapes indicate whether they are essential in animals. In (A), y=x is shown as a dashed line, in all other plots dashed lines for x=0 and y=0 are shown for clarity. In (A), Spearman's ρ = 0.94, p = $6 \times 10^{-6}$. Spearman correlations for (B), (C), and (D) are not significant (p = 0.8, 0.8, and 0.2, respectively).

The online version of this article includes the following figure supplement(s) for figure 2:

**Figure supplement 1.** Phylostratigraphy slopes are not significantly correlated with hydrophobicity, as measured by 1-relative solvent accessibility (RSA) (*Tien et al., 2013*).

specific pfams have 3.5% cysteine, as opposed to 2.6% in animal-specific pfams, and 2.1% in ancient pfams.

Photosynthesis likely creates greater oxidative stress in plants than in animals, which is likely to disrupt disulfide bonds. But in the training data used by IUPred, cysteine residues are mostly in stable disulfide bonds (*Mészáros et al., 2018*), so IUPred scores cysteines as highly order promoting. If unpaired cysteines are common in plants, this might obscure a trend in ISD. However, recalculating IUPred scores from sequences from which we had excised cysteine resulted in very little change in our ISD results (*Figure 1—figure supplement 4*).

In young animal pfams, the two amino acids that are most enriched are serine and proline. These are two of the amino acids previously identified as most responsible for differences between eukaryotes and prokaryotes, due in particular to the greater quantity of 'linker' regions (protein sequences between pfams) of eukaryotic proteins (*Basile et al., 2019*). Our results suggest instead that there is nothing special about linker regions. Linker regions might simply be young sequences that happen not to have been annotated as a pfam domain. We find that not just linkers, but also young annotated pfam domains in animals have a high percentage composition of serine and proline. This is perhaps unsurprising; the simple categorization of sequences as either 'pfam' or 'linker' in genome annotation is artificial and might introduce a variety of biases.

Because animal species with more tyrosine kinases have less tyrosine, it has been argued that selection against deleterious tyrosine phosphorylation has driven a decline in tyrosine content in metazoa (*Tan et al., 2009*). However, evidence that selection drives tyrosine loss is limited (*Pandya et al., 2015*), and it has been argued that changes in GC content might explain trends in tyrosine (*Su et al., 2011*, although see *Tan et al., 2011*). Examining this question with a phylostratigraphy approach for the first time, we do not see evidence for tyrosine loss in animals; tyrosine has a relatively shallow phylostratigraphy slope that is indistinguishable from 0 (*Supplementary file 1*). This agrees with the findings of *Pandya et al., 2015*, who found no significant difference between the frequency spectra of alleles that removed or created tyrosine, suggesting that tyrosine is neither strongly selected for nor against, with trends in tyrosine being modest compared with those of other amino acids. There is, however, more tyrosine in younger ancient pfams than in the oldest ancient pfams, compatible with an ancient rather than recent process of tyrosine loss (see *Supplementary file 1*).

To assess whether animal-specific trends also reflect amino acid availability, we examined essential amino acids (i.e. amino acids that plants but not animals can synthesize [*Guedes et al., 2011*]), expecting them to be rarer in young animal domains. While this is the case (with six of nine of their phylostratigraphy slopes being positive; *Figure 2*), the difference is not significant (Binomial test with 50% chance of a positive slope, one-tailed p = 0.3).

Instead, the ISD score returned by IUPred seems to best summarize animal-specific trends in overall sequence hydrophobicity. We do not find differences among amino acids in hydrophobicity scores such as relative solvent accessibility (RSA) (*Tien et al., 2013*). While we do see a tendency for hydrophobic amino acids to have positive phylostratigraphy slopes among animal pfams and negative slopes among plant pfams (amino acids are colored by RSA in *Figures 2* and *3*), this correlation between RSA and phylostratigraphy slope is not significant for any of the lineage- or phylostrata-specific subsets of the data set shown in *Figure 2* (see *Figure 2—figure supplement 1*).

What is more, if hydrophobicity was the main determinant of phylostratigraphy slopes, we would expect amino acid composition to evolve differently in a hydrophobic membrane environment than in the cytosol. However, amino acid slopes are highly correlated between transmembrane and non-transmembrane ancient pfams (*Figure 3A*). Clearly, trends among ancient domains are not primarily driven by hydrophobicity. Amino acid slopes are also weakly, albeit not significantly, correlated between transmembrane and non-transmembrane pfams in animals (*Figure 3B*). Breaking this animal correlation is leucine, a clear outlier in that it is enriched in young transmembrane pfams but depleted in young non-transmembrane pfams. Leucine also shows the same pattern to some degree among ancient pfams. There is very little power to detect a correlation in plants, should it exist (*Figure 3C*).

There has been concern that phylostratigraphy slopes could be driven by homology detection bias (*Moyers and Zhang, 2015*, *Moyers and Zhang, 2016*; *Wilson et al., 2017*). In this case, amino acids that are more changeable, as assessed by the changeability scores of *Tourasse and Li, 2000*, making homology more difficult to detect, should be over-represented in young genes. However,

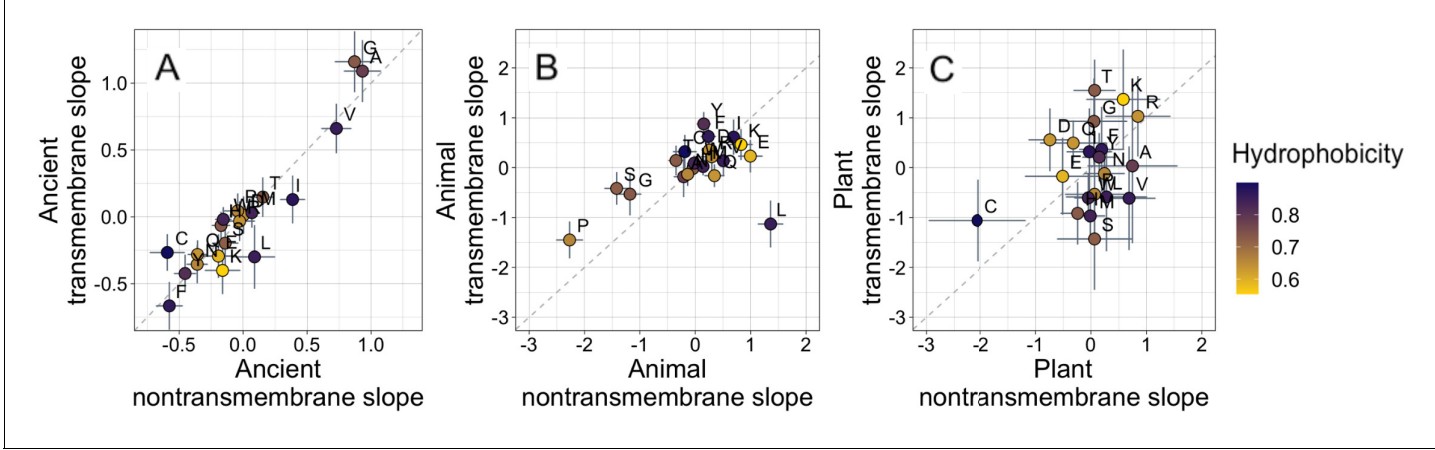

**Figure 3.** Ancient domains exhibit similar amino acid trends, whether transmembrane or non-transmembrane. Phylostratigraphy slopes are in units of percentage point change in composition per billion years. Taxonomic and temporal sub sets of the data are as the same as in *Figure 2*. Lines indicate the standard errors on the slopes. Points are color-coded by their hydrophobicity, as measured by 1-relative solvent accessibility (RSA) (*Tien et al., 2013*), such that buried, hydrophobic amino acids are dark blue, and exposed, hydrophilic amino acids are yellow. In all plots, y=x is shown as a dashed line. Congruence is strong for ancient pfams in (**A**) (Spearman's ρ = 0.83, p = 5 × 10⁻⁶), weakly significant for animal-specific pfams in (**B**) (p = 0.06) where transmembrane trends tend to be weaker and leucine is an outlier, and not detectable in our weakly powered plant-specific data set in (**C**) (p = 0.3).

The online version of this article includes the following figure supplement(s) for figure 3:

**Figure supplement 1.** Phylostratigraphy slopes are not significantly correlated (after correction for multiple testing) with relative amino acid changeability.

**Figure supplement 2.** Phylostratigraphy slopes are not significantly correlated with the amino acid flux estimates of *Jordan et al., 2005*.

we find if anything the opposite: more changeable amino acids are (slightly) enriched rather than depleted among the oldest of ancient pfams (*Figure 3—figure supplement 1*, Spearman's ρ = 0.50, p = 0.02 for nontransmembrane pfams, and ρ = 0.40, p = 0.08 for transmembrane ancient pfams). Our results are therefore not driven by homology detection bias even at extremely long timescales. There is no correlation between recent animal (p = 0.57) or recent plant (p = 0.7) phylostratigraphy slopes and amino acid changeability, and therefore no evidence that these results are affected by homology detection bias either. Furthermore, we note that homology detection bias is expected to create similar patterns for all taxa. The fact that the strength and even direction of amino acid trends can be different for different taxa is itself also evidence against a strong role for homology detection bias.

Given the striking similarity in ancient trends between transmembrane and non-transmembrane domains, we note that *Jordan et al., 2005* claim ongoing trends related to the origin of the genetic code. Specifically, they claim that the amino acids incorporated earliest into the genetic code (*Trifonov, 2000*) tend to be lost, while more recently acquired amino acids tend to be gained. *Jordan et al., 2005* used a parsimony-based method of ancestral sequence reconstruction to detect trends in the amino acid composition of existing protein-coding sequences over time. The method has, however, been much criticized (*Goldstein and Pollock, 2006*; *Hurst et al., 2006*; *McDonald, 2006*).

Our amino acid phylostratigraphy slopes do not correlate with the flux values of *Jordan et al., 2005* (*Figure 3—figure supplement 2*). However, we find that among ancient domains, the very oldest domains are enriched in the amino acids that were hypothesized to have been recruited into the genome first, both in nontransmembrane domains (Spearman's ρ = −0.62, p = 0.004), and in transmembrane domains (Spearman's ρ = −0.55, p = 0.01). This suggests that amino acid availabilities at different stages of life could have affected the composition of proteins born at those stages, in an effect that persists to this day. Specifically, it suggests that amino acids recruited later into the genetic code increased in abundance only slowly, creating long-term trends in amino acid availability among different ancient origin dates.

The order of recruitment ranking is taken from *Trifonov, 2000*, who estimated a possible consensus chronology from the expectations of 40 amino acid ranking criteria, including such diverse factors as results from experiments into primordial conditions, amino acid complexity, and thermostability. However, 3 of the 40 criteria were based on the amino acid compositions of protein assemblages. Therefore, to ensure non-circularity, we recalculated the consensus order, excluding these criteria and calculating the mean rank, as in *Trifonov, 2000* prior to their proposed second step of smoothing by a filtering procedure. This recalculation had very little effect on the order of amino acids, and our correlation in ancient nontransmembrane pfams remains significant (*Figure 4A*, Spearman's $\rho = -0.58$, $p = 0.008$), although the correlation for ancient transmembrane pfams is no longer on its own robust to multiple testing (*Figure 4B*, Spearman's $\rho = -0.47$, $p = 0.04$).

The order of recruitment ranking is highly speculative, and correlations with it could be driven by a single correlated amino acid physiochemical property. Specifically, amino acids with very simple structures (A, G, and V) have steeper positive phylostratigraphy slopes than any other amino acids, and this alone may drive our observed correlation between phylostratigraphy slopes and order of recruitment. We note that order of recruitment is not significantly correlated with phylostratigraphy slopes in more recent lineages (*Figure 4—figure supplement 1*). This is in agreement with the

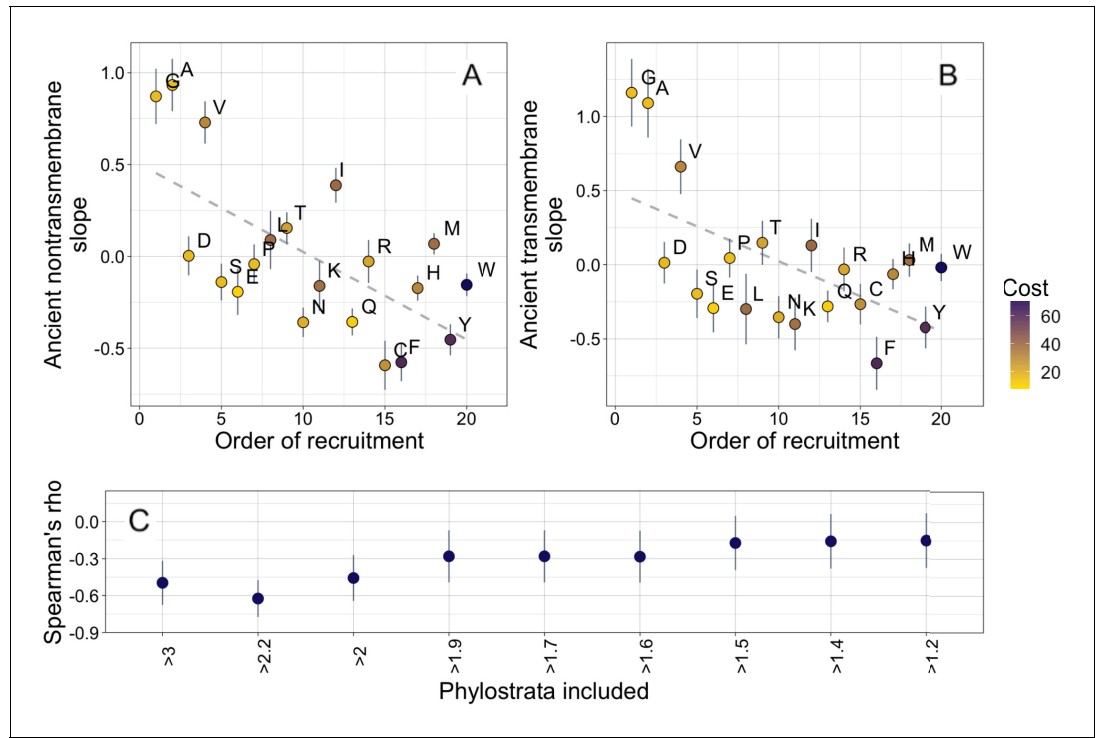

**Figure 4.** Ancient amino acid phylostratigraphy slopes reflect the order of recruitment of amino acids into the genetic code. Phylostratigraphy slopes for non-transmembrane (**A**) and transmembrane (**B**) pfams are in units of percentage points of composition per billion years, with lines indicating the standard errors on the slopes. Phylostratigraphy slopes for ancient pfams are calculated over all lineages, and include all pfams over 2101 MY. Consensus order is modified from *Trifonov, 2000* (see description in Results) and is given as rank data, such that the first amino acid to be recruited is given a rank of 1. Regression slopes are shown as dashed lines. Late-recruited amino acids are rare in the most ancient non-transmembrane pfams (**A**) (Spearman's $\rho = -0.58$, $p = 0.008$) and transmembrane pfams (**B**) (Spearman's $\rho = -0.47$, $p = 0.04$). Points are color-coded by their metabolic costliness to produce (measured as the number of high energy phosphate bonds required for synthesis, plus the energy lost due to the precursors used in the synthesis), as estimated for aerobic conditions in yeast (*Raiford et al., 2008*), such that the most costly amino acids are blue, and the least costly are yellow. (**C**) The correlation (Spearman's $\rho$) between amino acid phylostratigraphy slope and order of recruitment over different subsets of our data set. X-axis labels indicate the minimum age of pfams included, in billion years. Lines are the standard errors of the Spearman's $\rho$ values, calculated using the Fisher transformation (*Fisher, 1915*).

The online version of this article includes the following figure supplement(s) for figure 4:

**Figure supplement 1.** Order of amino acid recruitment does not affect domain composition in more recent lineages.

**Figure supplement 2.** Phylostratigraphy slopes are not significantly correlated (after correction for multiple testing) to the cost of production of amino acids (aerobic metabolic cost, as estimated in yeast [*Raiford et al., 2008*]).

broader hypothesis of *Jordan et al., 2005* that the order of recruitment shaped early amino acid availabilities and hence protein compositions, but is hard to reconcile with alternative explanations in which amino acid simplicity is directly causally responsible.

Our interpretation of these results is that amino acids that were introduced late into the genetic code remained relatively rare for a prolonged period of time, well after the genetic code was complete. To determine for how long, we ask when the correlation between phylostratigraphy slope and order of recruitment is strongest (*Figure 4C*). The strongest correlation is with slope over the three oldest phylostrata in our data set. These span the pfams found in last universal common ancestor (LUCA), through to pfams present in all extant eukaryotic but no prokaryotic lineages. The strength and significance of the correlation between order of recruitment and phylostratigraphy slope decrease if younger phylostrata are included, suggesting that from the LECA onward, the order of recruitment into the genetic code no longer shaped amino acid availability.

We do not believe our order of recruitment results to be a byproduct of a correlation with amino acid costliness. This is because there is no more than marginal significance for correlations between phylostratigraphy slope and the metabolic cost of amino acid production in yeast (*Figure 4—figure supplement 2*; amino acids are color-coded by costliness in *Figure 4*; *Raiford et al., 2008*).

## Trends in clustering

Finally, we consider whether there are any trends in amino acid order. Specifically, a temporal trend in the value of a 'clustering' metric has previously been reported for mouse gene families (*Foy et al., 2019*). This clustering metric calculates the degree to which hydrophobic amino acids tend to lie close together along the primary sequence, normalized to ensure that values do not depend on length or amino acid frequencies (*Irbäck et al., 1996*) (see Materials and methods for a full technical description). Not only is there no causal link by which amino acid composition affects clustering, there is also no significant correlation (Pearson's p = 0.06 across non-transmembrane pfams) We confirm, using our considerably larger data set, that young pfams (*Figure 5A*, slope = −0.037, $R^2$ = 0.028, p = $1 \times 10^{-50}$), and young proteins (*Figure 5—figure supplement 1A*; slope = −0.031, $R^2$ = 0.033, p = $5 \times 10^{-132}$) have more clustering of their hydrophobic amino acids. In *Figure 5—figure supplement 1B*, we show that our improved methods for assigning gene age result in a steeper slope, that is, a larger effect size, for mouse genes than that reported by *Foy et al., 2019*: −0.056 instead of −0.045. We note that while the proportion of variance in clustering that is explained by age is small, this is driven at least in part by large denominator variance due by domain function or random variation.

Unlike the trend in ISD, the trend in clustering with age is remarkably consistent across age categories (*Figure 5—figure supplement 2*, *Figure 5B*: slope = −0.067, p = 0.0005 for animal-specific domains and slope = −0.025, p = 0.008 for ancient domains). While the confidence intervals increase for subsets that contain less data (such as the set of young plant pfams, for which we have little power), they always overlap the clustering slope calculated over all lineages and all phylostrata. This suggests that trends in clustering may be universal. If new sequences were born with similar clustering values, this would imply that the dispersion of hydrophobic amino acids change at an approximately constant rate over the whole of evolutionary time, with the observed trend resulting from the fact that the change has had more time to take place in older sequences. Under this interpretation, the data is also compatible with a slower contemporary rate of change of clustering in ancient pfams.

## Discussion

We have shown that in our data set of eukaryote genomes, there is a universal trend for young genes and pfam domains to have clustered hydrophobic amino acid residues, while old domains and genes have more evenly interspersed hydrophobic amino acids. In contrast, trends in amino acid content depend on which lineage is examined. Only animal trends are driven by young protein-coding sequences having high ISD. Ancient trends, from LUCA to the LECA, are driven by the order of recruitment of amino acids into the genetic code. We have less power to decipher plant trends, although we note a particularly strong excess of cysteine, especially relative to the generally low levels at which abundant cellular cysteine is incorporated into plant proteins as a whole.

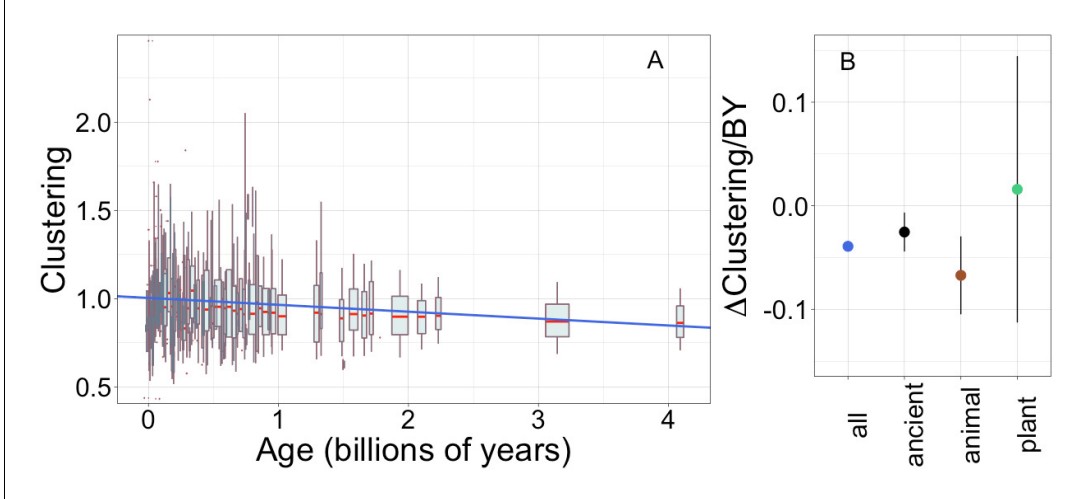

**Figure 5.** Young domains have more clustered hydrophobic amino acids (**A**), and the trend in clustering with age is consistent across time and taxonomic groups (**B**). n (number of pfams) = 8002, 3100, 2456, and 183 for all, ancient, animal, and plant groups, respectively. Clustering has an expected value of 1 for randomly distributed amino acids. Results are shown for non-transmembrane pfams. (**A**) Each data point consists of the average across all homologous instances of a pfam, across all species in which the pfam occurs. Phylostratigraphy assigns these to age classes, dated using timetree. To visualize our results, every age class is represented by a weighted box plot, where the width of the plot indicates the number of pfams in that age class. The median is shown in red, with the boxes representing upper and lower quartiles (the 75th and 25th percentile), and the whiskers indicating 9 and 91 quantiles. For age classes with only a single pfam, values are presented as red dots. For clarity of presentation our plots do not show outliers, although these are included in our linear regression models. The blue line is the linear regression slope calculated over all lineages. Slope = −0.039, $R^2$ = 0.030, p = 3 × $10^{-54}$. Slope represents the average decrease in clustering score per billion years. (**B**) Phylostratigraphy slopes (in units of change in clustering per billion years) for pfams are calculated over different phylostrata subsets of the data (**Figure 5—figure supplement 2**), and shown with their 95% confidence intervals. The left-most point corresponds to the regression slope in (**A**).

The online version of this article includes the following figure supplement(s) for figure 5:

**Figure supplement 1.** Hydrophobic clustering of complete genes depends on age, as previously reported by **Foy et al., 2019**.

**Figure supplement 2.** Young animal domains have more clustered hydrophobic amino acids, continuing the trend among ancient domains that can be seen in both animals and plants.

The effect sizes of sequence age on amino acid frequencies are small. The question is whether they are biologically important. **Kosinski et al., 2020** estimated the marginal effects of substituting one amino acid on the fitness associated with expressing a random peptide in *E. coli*. Encouragingly, they found that those marginal effects correlate with our animal phylostratigraphy slopes, suggesting that our results are indeed driven by the forces that facilitate gene birth. For example, a single switch of one proline in a 50-mer changes genotype frequency by 3% over the course of one cycle of experimental evolution. We can think of that switch as a two percentage point change in proline frequency, that is, of the order of change we see per billion years. While this fitness effect is small, it nevertheless has the potential to be biologically relevant.

Our analysis relies on homology detection of pfam domains. It is important to consider whether systematic errors in homology detection could bias our results. Our improved hmmer-based homology detection methodology should reduce the false negative rate, and thus the scope for homology detection bias. But both our ISD and our clustering trends are stronger rather than weaker than those previously reported using blastp-based methods (**Foy et al., 2019**; **Wilson et al., 2017**). Homology detection bias also cannot explain why trends in ISD or amino acid composition are lineage-specific, nor the absence of correlation with amino acids' evolutionary changeabilities. Additionally, old sequences are expected to be most affected by homology detection bias (**Moyers and Zhang, 2017**), but it is more recent animal domains that drive the ISD result. Overall, we find substantial evidence contradicting the suggestion that homology detection bias drives trends.

The rejection of homology detection bias does not imply that distant homology is always successfully detected, but rather that errors in age assignment might be primarily random rather than systematic. Evidence is accumulating that de novo gene birth is not, as once thought, rare (or even non-existent) after the origin of life. A recent analysis exploited synteny to find that most

taxonomically restricted genes are not the product of divergence, but have instead emerged de novo (*Vakirlis et al., 2020*). Another recent study excluded genes with insufficient power to detect ancient homology, and similarly found that evidence for de novo origin persisted in a substantial number of cases (*Weisman et al., 2020*). To get correct trends given random failures in homology detection, all we need is for apparent ages to be correlated with true ages.

Sequences can also be assigned as older than they really are, when they are incorrectly classified as present in distantly related species (false positives), due either to overly permissive homology detection cutoffs, or to contamination (see, for example, *Longo et al., 2011*; *Merchant et al., 2014*). In addition, some domains may be present in species due to horizontal gene transfer, which is problematic for the purpose of our analysis as it means that our annotation based on presence in a species does not reflect the evolutionary history of that domain (*Liebeskind et al., 2016*; *Ravenhall et al., 2015*). We used a number of filters to remove false positives and likely products of HGT from our analysis (see Materials and methods). False positives are not expected to be subject to systematic error with respect to ISD or clustering (*Moyers and Zhang, 2018*). In agreement with this, we found that these steps reduced the amount of 'noise' in our data set, making the trends we observed stronger with each improved iteration of our quality control.

Our results are consistent with the hypothesis that selection acts to reduce the aggregation propensity of proteins. Polypeptide chains are intrinsically prone to forming aggregates that are toxic to cells (*Chiti and Dobson, 2017*). Young animal genes avoid aggregation by being on average more disordered and hydrophilic (*Wilson et al., 2017*), with their few hydrophobic residues tending to cluster together along the protein chain (*Foy et al., 2019*). *Kosinski et al., 2020* directly showed that high ISD tends to make random de novo sequences less harmful when expressed in *E. coli*. However, in order to be retained, sequences must also consistently serve a function within their host cells; this often requires them to fold, which is promoted by higher hydrophobicity. It is possible that pfam domains eventually reach a stable level of disorder, a balance between selection for folding, selection against misfolding, and mutational pressure. This may be why we do not see trends in ISD in ancient domains.

Frustration between the two aims of selection might be resolved by sequences evolving more sophisticated strategies for folding while avoiding aggregation. One such strategy appears to be the ordering of amino acid residues (*Bertram and Masel, 2020*; *Foy et al., 2019*), with older, more evolved sequences having more evenly dispersed hydrophobic amino acid residues. Remarkably, we find that the trend in clustering is common to all lineages (animal, plant, and pre-LECA) included in our analysis, with no evidence for saturation at a stable level of clustering. Hydrophobic amino acids are more evenly dispersed than would be expected from chance in old proteins, ruling out a trend toward a steady state set by mutation (*Foy et al., 2019*), The trend lowers aggregation propensity compared to the expected value based on their amino acid compositions (*Foy et al., 2019*), which is compatible with the trend being due to selection having had more time to decrease the clustering of hydrophobic amino acids. If so, then even over vast amounts of evolutionary time, selection continues to reduce levels of clustering in protein domains rather than reaching a steady state. This may be because decreasing clustering is difficult. For a sequence to reduce levels of hydrophobic amino acid clustering while maintaining a similar amino acid composition, many substitutions are needed on a rugged fitness landscape (*Bertram and Masel, 2020*).

Disordered regions can contain 'sticky' amino acids at protein–protein interaction surfaces (*Levy et al., 2012*), whose hydrophobicity within a hydrophilic region will decrease clustering. More abundant proteins have fewer sticky residues within their disordered regions (*Dubreuil et al., 2019*). Older domains generally have higher protein abundance (*Carvunis et al., 2012*), which should cause lower stickiness and higher clustering, not the lower clustering seen in our results. Our clustering trend is therefore not driven by trends in protein 'stickiness', which is a possible contributor to aggregation propensity.

Protein length is another hidden factor that might contribute to the trends we see. It is a priori likely that younger proteins are shorter (*Van Oss and Carvunis, 2019*) with a correspondingly higher surface to volume ratio, and thus a proportionately smaller hydrophobic core and higher ISD score. However, this cannot explain our results, because while younger animal domains do tend to be shorter, shorter animal domains are actually less disordered ($R^2 = 0.0025$, $p = 0.007$). The reasons for this are unclear; length and disorder are, as expected from surface to volume considerations, negatively correlated in ancient domains and in recent plant domains.

Differences between taxa are surprising, with young plant domains having different trends in amino acid frequencies from young animal domains, and not sharing their preference for high ISD. Plant domains as a whole are more ordered, with ancient domains in plants also having lower ISD in plants than the same domains in animals (although the difference in ISD is most pronounced in lineage-specific domains). We speculate that plants are simply better at coping with proteostasis than other taxonomic groups, using mechanisms other than high ISD to protect themselves from protein misfolding and aggregation. Plant proteomes are rich in predicted amyloidogenic regions (*Antonets and Nizhnikov, 2017a*), and yet there is a paucity of evidence for aggregate formation in plants under normal conditions (*Antonets and Nizhnikov, 2017b*). This is perhaps due in part to the presence of molecules such as polyphenols, which inhibit aggregation (*Velander et al., 2017*). The effectiveness of plant molecules at inhibiting aggregation has also been related to plant longevity (*Mohammad-Beigi et al., 2019*). Young, ordered plant proteins may therefore be under weaker selection against aggregation than the same protein would be in an animal. Our data set does not have enough plant genomes to give us power to detect whether selection acts on clustering in plants.

The degree of cysteine enrichment in young plant domains is striking, standing out despite the low power of our plant analyses. We hypothesize that young plant domains might be cysteine-rich because cysteine is relatively available in the cell. Producing cysteine is the final step in the assimilation of sulfur by plants. Additionally, cysteine production enables the detoxification of reactive oxygen species produced during photosynthesis and is essential to chloroplast function (*Bermúdez et al., 2010*). In the cytosol, levels of cysteine appear to play a role in maintaining redox capacity and in pathogen resistance (*Romero et al., 2014*). Despite the abundance of cysteine in the cytosol, its incorporation into protein sequences, while inexpensive in terms of raw materials, might be deleterious in terms of risk of oxidative damage and unstable disulfide bonds. This might explain why cysteine is progressively lost after high abundance at the time of birth. After cysteine, the two amino acids that are most enriched in young plant pfams are glutamic acid and aspartic acid, E and D. These are the most abundant amino acids in plants as a whole (*Kumar et al., 2017*) and may therefore also be enriched in young plant domains due to their high availability.

Genomic GC content might determine the amino acid composition of young genes, causing high %GC taxa to spawn more hydrophilic and thus disordered young genes (*Ángyán et al., 2012*). Our animal vs. plant data have 50% and 47% GC, respectively (Welch's t-test, p = 0.0001), which could account for several percentage points of difference in ISD (*Ángyán et al., 2012*), but not for the much larger difference in young domain ISD seen in *Figure 1A*. We note that GC content evolves quickly between species and is variable on a finer taxonomic scale than is considered in this study. For example, monocot and dicot plants have a greater average difference in GC content than do animals and plants (*Li and Du, 2014*; *Romiguier et al., 2010*; *Šmarda et al., 2014*). Differences in GC content can also not explain the relative excess of cysteine in young plant domains. From inspection of the genetic code, we expect cysteine to easily accommodate any genomic %GC between 1/3 and 2/3, and this is confirmed by its prevalence as a function of species %GC (*Li et al., 2015*).

Our data are consistent with a hypothesis of selection acting in the usual way, that is, that proteins born de novo go on to discover, through mutation, more sophisticated strategies for folding while avoiding aggregation and/or misfolding. A second, not mutually exclusive possibility is that the trends we observe are due to differential retention of pfam domains and genes, rather than descent with modification. Under this hypothesis, newborn sequences are diverse, and sequences with favored properties are lost less often during subsequent long-term evolution. The relative contribution of these two levels of selection could be investigated if quality control could be made stringent enough to obtain quantitatively reliable loss rates and diversification rates for pfam domains.

In conclusion, trends in the evolution of amino acid composition show surprising differences between taxonomic groups and over different spans of time. Amino acid availability at time of de novo birth has a strong effect on domain composition. Even after billions of years of evolution, ancient domains remain influenced by which amino acids were more abundant as the genetic code was being formed. De novo birth and subsequent trends in plants may also be shaped by amino acid availabilities. In young animal domains, amino acid composition trends may be due to selection for high ISD. However, the inferred ancestors of eukaryote protein sequences show a universal trend toward reducing their levels of hydrophobic clustering with age, no matter their amino acid composition, presumably in an attempt to find a balance between folding and misfolding. The only claim in

evolutionary biology that is close to comparable in timescale is the increase in body size in some taxa over the 3.5 billion-year history of life (*Cope, 1885*; *Heim et al., 2015*; *Payne et al., 2009*).

## Materials and methods

### Data compilation

We compiled a data set of whole-genome sequences with a sequencing status of 'complete' from the GOLD database (*Mukherjee et al., 2019*, accessed August 7, 2018), resulting in a list of 1138 unique species. We then downloaded species from the Ensembl Biomart interface databases (*Kinsella et al., 2011*; *Zerbino et al., 2018*, fungi V40, plant V40, metazoan V40, protest v40, and main V93, accessed between July 31, 2018 and September 12, 2018). Of the resulting 306 species, 19 were not found in the GOLD species list and thus were excluded. We next searched the NCBI RefSeq repository (*O'Leary et al., 2016*, accessed between September 27, 2018 and October 26, 2018) for the remaining GOLD species, excluding archaeal, bacterial, and viral genomes due to lack of phylogenetic resolution. Of the resulting 948 species, 344 had been annotated, were in the GOLD species list, and were not also in Ensembl Biomart.

We further required all species in our data set to be present in TimeTree (*Hedges et al., 2006*), eliminating 163. Duplicates were removed (seven species), as were species using an alternative coding alphabet (two species), or species for which there were genome annotation quality issues (such that we were able to identify unusually many more pfams from blast searches of intergenic regions than were present in the currently annotated genome, and/or encountered technical problems such as many annotated genes being rich in stop codons) (four species), resulting in a final data set of coding sequences of 455 unique species.

Protist genomes were retained, but used only as outgroups for dating (for a review, see *Eme et al., 2014*), while full protein properties were calculated for a core set of 435 animal, plant, and fungal species. Protein-coding sequences of the 455 species in the curated list were then downloaded from the previously listed Ensembl repositories from their respective FTP sites, or from NCBI's FTP RefSeq repository. For a full set of species used in the analysis, see *Supplementary file 2*. Following genome acquisition, mitochondrial and chloroplast genes were removed from the data set.

Finally, genes often have multiple annotated transcripts/proteins, resulting in non-independent datapoints. We chose a single transcript to represent each gene, specifically the closest homolog to the most closely related sister species in our data set, identified using reciprocal Blastp (*Altschul et al., 1990*). If a gene failed to produce any hits below an e-value cutoff of $10^{-3}$, the longest gene transcript was chosen instead.

### Domain annotation

Ensembl provided Pfam annotations for all protein transcripts, which are based on InterProScan with default parameters. These annotations were manually downloaded from the BioMart web interface and paired with their corresponding protein using Ensembl's unique transcript identifiers. To make our data set internally consistent, we replicated Ensembl's annotation methodology by processing all NCBI sequences with InterProScan (*Jones et al., 2014*, v.5.20–69.0 downloaded June 20, 2018) using default parameters. All sequences without an associated Pfam were discarded from our analyses.

### Domain filtering

Because a well-resolved tree is crucial for correct age assignment, we focus on eukaryotic genes. Contamination is a major problem in gene annotation (*Kryukov and Imanishi, 2016*; *Merchant et al., 2014*; *Salzberg, 2017*), so we filtered our data set for likely contaminants, in addition to horizontally transferred genes. While horizontally transferred genes are not contaminants, their unusual form of inheritance means is likely to produce incorrect estimates of gene ages. The same applies to the unique evolutionary history of organelles.

Out of 17,929 available pfams, we first excluded 4913 pfams that appeared in our eukaryotic data set despite being annotated as occurring in prokaryotes, but not eukaryotes (as annotated in the tree file 'pfamA_species_tree.txt', accessed on 17 May, 2019 from https://pfam.xfam.org/).

Next we excluded pfams via keyword search. Of the 5076 pfams annotated as exclusive to eukaryotes, 218 pfams that contained the terms 'mitochondria' or 'chloroplast' in the interproscan abstract were excluded, leaving 4858 pfams. Of these eukaryote-specific pfams, we further excluded 561 pfams for which there was no interpro abstract available, leaving 4297 pfams out of the 4858.

Of the 7773 pfams annotated as occurring both in eukaryotes and in bacteria, archaea, or viruses, we excluded 2064 that contained the strings 'viral', 'virus', 'bacter*', 'capsid', 'bacillus', 'Pilus', 'Pilin', 'mitochondria', or 'chloroplast' unless they also contained one of the terms 'eukary*', 'vertebrat', 'fung*', 'metazoa', 'plant', 'mammal', 'insect', 'yeast', 'human', 'all organisms', 'human', 'antibod', and 'immune', leaving 5709 pfams out of the 7773. Of these ancient pfams shared with prokaryotes, we further excluded 1112 pfams without an available interpro abstract, leaving 4597 pfams.

The 167 pfams that did not have species annotation information available in the pfam database (https://pfam.xfam.org/) were removed from the data set, resulting in an overall total of 8894 pfams.

The phylogenetic distribution of some remaining pfams appeared indistinguishable from chance contamination. To formalize this criterion, we compared the number of losses inferred by Dollo parsimony to that occurring under the hypothesis that all hits are random contamination, performing this test for all pfams reported in less than half the total number of species. For each possible number of species containing the pfam, we simulated a distribution of the inferred number of losses when that number of species was chosen at random. We then used the mean and variance for the resulting distribution to calculate z-scores for the number of inferred losses from the actual data on each pfam, and rejected the pfam if z-scores $> -2$, a total of 688 pfams. This resulted in a final set of 8209 pfams.

We also remove all genes that contain a pfam excluded as described above.

## Age assignment

### Pfams

Pfam domains were assigned a date of evolutionary origin using TimeTree, using the 'build a timetree' option. This option returns the mean divergence times calculated over all published molecular time estimates. We assumed each pfam originated halfway between the node of the most recent common ancestor of all species in which the pfam originated, and the node prior to that. To add additional temporal resolution to the oldest domains, we identified any Pfams in our data set likely to have been present in the first eukaryotic common ancestor, defined as those pfams present both in non-protist eukaryotes and in the protist group 'Excavata' (species included in our data set are: *Naegleria gruberi, Giardia intestinalis, Giardia lamblia, Spironucleus salmonicida, Angomonas deanei, Bodo saltans, Leishmania braziliensis, Leishmania donovani, Leishmania infantum, Leishmania major, Leishmania mexicana, Leishmania panamensis, Leptomonas pyrrhocoris, Leptomonas seymouri, Perkinsela* sp., *Strigomonas culicis, Trypanosoma brucei, Trypanosoma cruzi, Trypanosoma rangeli, Trichomonas vaginalis*), which is considered a possible outgroup to the other eukaryotes, and with an estimated date of 2230 MYA (*Hedges et al., 2001*; *Hedges et al., 2006*; *Parfrey et al., 2011*).

Beyond TimeTree, pfams inferred to have been present in the LUCA (*Weiss et al., 2016*) were dated as 4090 MY old. For those pfams that are found in eukaryotes and bacteria, but not archaea, and also those pfams that are found in eukaryotes and archaea, but not bacteria, that is, for those that emerged after LUCA but prior to the emergence of eukaryotes, we assigned them the age of 3145 MY, as the halfway point between the emergence of eukaryotes and LUCA. Dates of these ancient events are imprecise. Recent work has also highlighted the difficulty in identifying domains present in LUCA (*Berkemer and McGlynn, 2020*).

## Full genes

We dated each gene according to the age of its oldest Pfam. Pfams represent highly conserved sequences, so it is unlikely that genes would be categorized as older based on non-Pfam sequences. Note that while different parts of a gene have different ages, here we classify a protein's age based on the age of its oldest pfam. We excluded genes without any annotated pfams from this analysis, which helps to remove sequences that are not truly genic, but may result in an underestimate of

disorder in our data set as a whole if it is the youngest, most disordered genes that are least likely to have annotated domains.

## Homology groups

Homologous sequences share a common evolutionary origin, and therefore treating them as independent datapoints is a form of pseudoreplication (*Wilson et al., 2017*). To correct for this, an average taken across each Pfam was treated as a single datapoint in our domain analyses.

For genes, this was achieved by grouping according to their oldest Pfam. For genes with multiple Pfams that are all equally the oldest, a cluster analysis was performed. We started by considering the frequency of the concurrence of two Pfams, *A* and *B*, $P(AB)$. If $P(AB| A\ or\ B) \geq 50\%$ given all protein-coding genes in our data set, then a link was established. Following all pairwise comparisons, Pfams were grouped together using single-link clustering, and each group was assigned a unique gene homology ID. An average across all genes sharing a gene homology ID was then treated as a single datapoint in our whole gene analyses.

The homology group files used in our analyses are available at https://doi.org/10.6084/m9.figshare.12037281.

## Metrics

### Transmembrane status

Genes and pfams were assigned as either transmembrane or non-transmembrane using the program tmhmm (*Krogh et al., 2001*), which predicts the number and position of transmembrane helices within a protein sequence. A gene was designated as transmembrane if it contained over 18 amino acids that are predicted to be in a transmembrane helix (*Krogh et al., 2001*). A pfam was designated as transmembrane if it overlapped with a predicted transmembrane helix by a minimum of 50% of the pfam length, or if a transmembrane helix overlapped with a pfam by a minimum of 50% of the helix length; 50% was chosen as an arbitrary albeit reasonable cutoff for the designation of transmembrane status.

## Disorder

Disorder predictions were made for each sequence using IUPred 2 (*Dosztányi et al., 2005a*; *Mészáros et al., 2018*), which assigns a score between 0 and 1 to each amino acid. Each protein's ISD score was calculated by averaging over the values of all amino acids. To determine the disorder of each Pfam, we averaged over only the relevant subset. In order to obtain results with interpretable units, we show untransformed results; however, results are qualitatively similar if data is transformed (using a Box-Cox transform with the optimal value of lambda to achieve normality) prior to analysis. We also recalculated IUPred scores after first removing cysteine residues from protein sequences.

## Amino acid composition

The fractional amino acid composition was found by counting the occurrences of each of the 20 amino acids in each protein or pfam and dividing by the length of the sequence. In order to obtain results with easily interpretable units, we show results for the untransformed proportions. Our qualitative results do not change if amino acid fractions are arcsine transformed (the most appropriate transform for proportions [*Sokal and Rohlf, 1995*]) prior to analysis (results not shown).

## Clustering

To determine clustering scores, we followed *Foy et al., 2019* and compared the variance of the hydrophobicity among blocks of $s$ = 6 consecutive amino acids to the mean hydrophobicity, to produce a normalized index of dispersion. If the length $L$ of a protein was not a multiple of 6, we took the average of all possible p = ($L$ modulo 6) frames, truncating the ends appropriately. The six most hydrophobic amino acids Leu, Ile, Val, Phe, Met, and Trp were scored as +1, and the remaining standard amino acids to −1, as in *Irbäck et al., 1996* and *Foy et al., 2019*. Non-standard amino acid residue abbreviations were also scored as +1 or −1, interpreted as follows: B as corresponding to D or N and hence −1; J as corresponding to either I or L and hence +1; Z as corresponding to either E or Q and hence −1; U as selenocysteine and O as pyrrolysine, both scored as −1.

For a sequence of length $N = \lfloor L/s \rfloor$ in frame $1 \le f \le p$, the scores for each block $k = 1,..., N/s$ of six amino acids were summed to $\sigma_{k,f}$ with the full truncated sequence summing in total to

$$M_f = \sum_{k=1}^{\frac{N}{s}} \sigma_{k,f}.$$

The normalized index of dispersion for a particular frame was then calculated as

$$\psi_f = \frac{s}{N} \sum_{k=1}^{N/s} \frac{1}{K} \left( \sigma_{k,f} - sM_f/N \right)^2$$

with the normalization factor

$$K = s\frac{N^2 - M_f^2}{N^2 - N} \left( 1 - \frac{s}{N} \right).$$

The total normalized index of dispersion was then calculated by taking the average over all possible frames

$$\psi = \frac{1}{p} \sum_{f=1}^{p} \psi_f.$$

Clustering scores were again analyzed as raw scores for ease of interpretation of slope units; however, results are qualitatively similar if the data are Box-Cox transformed prior to analysis, as for disorder.

## Statistical analyses

All statistical work in this research was performed using R. Our major results throughout the manuscript are based on 'phylostratigraphy slopes', which come from simple linear models of the effect of age on the particular protein metric of interest (i.e. ISD, percent amino acid composition, or clustering). Linear models include all homology groups. The function tidy from the broom package was used to convert model outputs to data frames for analysis and plotting, and to calculate 95% confidence intervals for regression slopes (*Robinson, 2014*). We used the base function lm to perform basic linear models. The lm function was also used to perform weighted least squares regression on variances. Plots were generated in R, using packages ggplot2 and gridExtra. The parametric boxplots were drawn using the quantile function in base R.

## Data availability

All scripts used in this work can be accessed at: https://github.com/MaselLab/ProteinEvolution copy archived at swh:1:rev:1c3b5dcc3dbce35acc1bbbfaafa29fc6398ecee8. Our raw data and homology files used in our analyses are available at https://doi.org/10.6084/m9.figshare.12037281.

## Acknowledgements

This work was supported by the John Templeton Foundation (60814) and the National Institutes of Health (GM-104040). The authors would also like to thank Heather Meyer for helpful discussions.

## Additional information

### Funding

| Funder | Grant reference number | Author |
| --- | --- | --- |
| John Templeton Foundation | 60814 | Joanna Masel |
| National Institutes of Health | GM-104040 | Joanna Masel |

The funders had no role in study design, data collection and interpretation, or the decision to submit the work for publication.

## Author contributions
Jennifer E James, Data curation, Software, Formal analysis, Investigation, Visualization, Methodology, Writing - original draft, Writing - review and editing; Sara M Willis, Data curation, Software, Formal analysis, Investigation, Visualization, Methodology, Writing - original draft; Paul G Nelson, Data curation, Formal analysis, Methodology, Writing - original draft; Catherine Weibel, Data curation, Software, Investigation, Writing - review and editing; Luke J Kosinski, Resources, Formal analysis, Methodology, Writing - review and editing; Joanna Masel, Conceptualization, Supervision, Funding acquisition, Writing - original draft, Project administration, Writing - review and editing

## Author ORCIDs
Jennifer E James (iD) https://orcid.org/0000-0003-0518-6783
Sara M Willis (iD) https://orcid.org/0000-0002-1605-6426
Catherine Weibel (iD) https://orcid.org/0000-0003-1837-5209
Luke J Kosinski (iD) https://orcid.org/0000-0002-8146-5955
Joanna Masel (iD) https://orcid.org/0000-0002-7398-2127

## Decision letter and Author response
Decision letter https://doi.org/10.7554/eLife.57347.sa1
Author response https://doi.org/10.7554/eLife.57347.sa2

# Additional files

## Supplementary files
• Supplementary file 1. Pfam amino acid frequency phylostratigraphy slopes, calculated over different subsets of the data.

• Supplementary file 2. The full set of species used in the analysis.

• Transparent reporting form

## Data availability
All scripts used in this work can be accessed at: https://github.com/MaselLab/ProteinEvolution (copy archived at https://archive.softwareheritage.org/swh:1:rev:1c3b5dcc3dbce35acc1bbbfaafa29f-c6398ecee8/). Our raw data, and homology files used in our analyses, are available at https://doi.org/10.6084/m9.figshare.12037281.

The following dataset was generated:

| Author(s) | Year | Dataset title | Dataset URL | Database and Identifier |
|---|---|---|---|---|
| James JE, Willis SM, Nelson PG, Weibel C, Kosinski LJ, Masel J | 2020 | Data from: Universal and taxon-specific trends in protein sequences as a function of age | https://doi.org/10.6084/m9.figshare.12037281 | figshare, 10.6084/m9.figshare.12037281 |

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
