## [Decision Letter]

**Acceptance summary:**

The authors present a comprehensive analysis of the composition of protein domains across millions of years of evolution to examine whether evolution is directing proteins towards specific regions of the available sequence space. One key finding is a reduction with time of hydrophobic clustering in proteins, consistent with evolution to reduce aggregation propensity as proteins age. The results presented show that although natural selection can change some protein features rapidly, other changes may be more difficult to achieve and can thus continue to be improved over long-term evolution.

**Decision letter after peer review:**

[Editors’ note: the authors submitted for reconsideration following the decision after peer review. What follows is the decision letter after the first round of review.]

Thank you for submitting your work entitled "Universal and taxon-specific trends in protein sequences as a function of age" for consideration by *eLife*. Your article has been reviewed by a Senior Editor, a Reviewing Editor, and two reviewers. The following individuals involved in review of your submission have agreed to reveal their identity: Taraneh Zarin (Reviewer #2).

Our decision has been reached after consultation between the reviewers. Based on these discussions and the individual reviews below, we regret to inform you that your work will not be considered further for publication in *eLife*.

Although the questions addressed in the manuscript are of potential interest to a broad community, the organization of the manuscript and some of the method descriptions make it difficult to know what are the specific questions being asked (Introduction) and how the data is compiled and analyzed (results). Some of the results appear to be significant but with very small effect sizes, which may require a more careful examination and interpretation. Since the study is focusing on protein domains, it would also be useful to define in the introduction what kind of protein domains are being analyzed. For many people, a protein domain has a tertiary structure, which means that by definition it is ordered. It is therefore not sure what a disorder predictor is predicting for such domains.

Reviewer #1:

Prior phylostratigraphic studies have reported correlations between various properties and gene age, where gene age is estimated from the last common ancestor of the species that harbor homologs of that gene. Failure to identify distant homologs is a source of error in gene age estimates. The prevalence of such errors, whether the error introduces systematic bias, and the impact of errors on phylostratigraphic inference is currently a subject of debate. This manuscript seeks to address these issues through the use of Pfam Hidden Markov models, instead of blastp searches, for homolog detection. The manuscript presents results that are relevant to two lines of inquiry:

– Does the use of Pfam HMMs reduce error in gene age estimates via greater sensitivity? Do the age estimates obtained with Pfam HMMs exhibit evidence of systematic bias that could lead to erroneous conclusions?

– What do gene age estimates obtained using Pfam HMMs tell us about the forces that govern the evolution of amino acid composition in proteins of different ages?

Both questions are important. However, there are substantial difficulties with the analysis and interpretation, as presented here, that must be addressed before is possible to assess the quality of the evidence and how well it supports the conclusions with respect to either question.

I) Lack of a formal hypothesis testing framework: In its current form, this manuscript does not address either of the above questions with sufficient formality and rigor to make a convincing case that the evidence supports a particular set of conclusions. To address this, the introduction should (1) summarize prior work, discuss open questions and unresolved controversies; (2) specifically state which hypotheses will be tested in the current study, with a discussion of the testable predictions that flow from these hypotheses; (3) describe how these predictions will be tested; and (4) what steps will be taken to eliminate confounding factors and rule out alternate hypotheses.

For the first question (mitigation of error due to more accurate homology identification), steps 1 and 2 are handled reasonably well. However, testable predictions are mentioned haphazardly in the results and confounding factors are not adequately addressed. The second question (what forces act on the amino acid composition encoded by genes as they age) is not formally stated as a target of inquiry, but introduced almost in passing in the middle of the manuscript.

II) Prior art: This manuscript states that "A key innovation is to use Pfam domains rather than BLASTp as our unit of homology". However, Pfam HMM models have been brought to bear on the question of homology detection and gene age estimates in a prior work that is not cited or discussed in the current manuscript: Jain et al., 2019. The authors will wish to read that article to determine to what extent it duplicates, contradicts or is complementary to the results in the current manuscript. In addition, they may find that the Jain et al. article offers useful ideas for methodological refinements and/or casts new light on how the results of the current manuscript should be interpreted. In particular, Jain et al., deal explicitly with lineage- and family-specific differences in evolutionary rates, topics that are not explored in depth in the current manuscript.

III) Methodology: The methodology in the current manuscript is not sufficiently well described to allow a reviewer to fully assess the results or another scientist to reproduce them. Technical terms are used without definition. Descriptions of how various quantities are actually calculated are lacking. Attention is given to statistical significance (p-values), but not to effect sizes. The presentation relies heavily on summary statistics and derived data, in ways that may obscure trends in the underlying raw data. Insufficient information is given about the data used, and it is not clear what data and supplementary information will be made available to the reader, other than the eight supplementary figures. Specific examples of these problems are given in the detailed comments below, but this is not an exhaustive list.

IV) Demonstrating that the evidence supports the conclusions: The manuscript contains a number of strongly word assertions without sufficient demonstration that these conclusions are in fact supported by the evidence. The logic underlying the assertions is not spelled out and alternate explanations are not ruled out or even discussed. The manuscript tends to conflate the description of an observation with the interpretation of what that observation means.

– In one example of these issues, the manuscript concludes that the use of Pfam HMMs resulted in improved detection accuracy. While this is highly plausible, the evidence for this conclusion is not rigorously demonstrated. The main evidence presented is that phylostratigraphy slopes for mouse genes are steeper when Pfam HMMs are used for homology detection, compared to blastp. However, the manuscript does not demonstrate that steeper slopes are incontrovertible evidence of more accurate homology detection. A comparison of the gene sets obtained using the two methods would provide a more convincing and direct assessment. Is there a decrease in false negatives without an increase in false positives? The properties of sequences in the two sets should also be examined for evidence of systematic bias.

The discussion of this issue exemplifies the tendency to confuse cause and effect. In the Results section, the statement "Our improved methodology increased the steepness of the relationship between gene ISD and gene age…" does not make a clear distinction between the observation (steeper slopes) and the inference (that steeper slopes are due to the new methodology and that the new methodology is an improvement). The inference must be demonstrated, not simply stated.

– In a second example, the argument is made that the observed relationships between amino acid composition and age are unaffected by systematic error in homology detection because different trends are observed in plant and animal protein domains and "homology detection bias is expected to create similar patterns for all taxa". No argument is presented to support the statement that "homology detection bias is expected to create similar patterns for all taxa." While this might hold under some conditions, it is not clear to me why this should be true in taxonomic lineages that differ substantially in GC content or evolutionary rates.

Even assuming that the "similar patterns for all taxa" prediction is valid, only one taxonomic comparison, plants versus animals, is offered as evidence. If subsets of animal species are compared, for example, are dissimilar patterns also observed? Further, could the different patterns in plants and animals be due to issues with the underlying data? The Materials and methods section describes a substantial effort to obtain high quality genomes. (This is one of the strongest sections in the Materials and methods.) Despite, and perhaps because of, this effort, the number of animal genomes exceeds the number of plant genomes by almost a factor of four. Domain discovery, modeling, and annotation in plant genomes lags substantially behind animal genomes, in part because of the relative dearth of proteomic data in plants. The fact that the slope for recent plant domains is positive and not statistically significant (p=0.1), as well as the large variance in plant slopes shown in Figure 1B, all suggest that there may be problems with the plant data that could be responsible for different trends.

The manuscript contains other, similar problems with interpretation of evidence, in addition to the two examples given above.

Reviewer #2:

James et al., comprehensively outline the challenges in phylostratigraphy and homology detection while carefully applying these methods to detect age-dependent trends in protein sequences. Their method improves signal for previously reported trends such as decreased hydrophobicity and increased hydrophobic clustering in young protein sequences. Interestingly, the authors find increased intrinsic structural disorder (ISD) in young animal domains, but not young plant domains. Their method also allows them to gain insights into changes in amino acid frequency with gene age. These results would be of general interest to *eLife* readers. The following could help clarify the scope of these results:

1) The authors should elaborate on the diversity of the taxa that are included. Although there are an impressive 435 species included in the analysis, there are only 5 non-plant and non-animal species, all of which are fungi. This is understandably due to quality control for data that are included in the analysis, but it's not clear if it's fair to use the term "universal" for the trends observed throughout the paper if they are based (mostly) on plants and animals.

2) The authors should comment on whether or not there are systematic differences in the plant vs. animal species that are included, and how that could affect the results of the study. For example, is the GC content of the included plant genomes a concern? There seem to be some reported differences in GC content of different plant species (monocots vs. dicots) [Kawabe and Miyashita, 2003; Li and Du, 2014, Šmarda et al., 2014] - are these species broadly sampled in this study? If not, this should be clearly stated.

[Editors’ note: further revisions were suggested prior to acceptance, as described below.]

Thank you for sending your article entitled "Universal and taxon-specific trends in protein sequences as a function of age" for peer review at *eLife*. Your article is being evaluated by Diethard Tautz as the Senior Editor, a Reviewing Editor, and two reviewers.

As you will see, the first reviewer was overall satisfied with the revisions, although some issues remained. The second reviewer raised important points, many of which are still valid for the latest version of the manuscript and concern some of the main conclusions.

Reviewer #2:

Having gone through the appeal letter and the current/revised version of the manuscript, I feel that the authors have addressed the reviewers' concerns, where possible and within the scope of the paper. The overall goals of the study are now more clearly defined, and the elaboration on the methods, data, and alternative explanations that could underlie the observations is helpful.

Reviewer #3:

Trends in summary statistics of proteins between plants and animals based on grouping age of origin show that bias in detecting homologous proteins is accounted for and not the strongest effect in results. Authors convincingly show (1) there are minor differences in degree of clustering of hydrophobic residues between proteins with ancient origins and more recent animal proteins; (2) animal and plant proteins with more recent origins have different degrees of predicted disorder; (3) Amino acid frequency trends are surprisingly consistent between ancient transmembrane and non-transmembrane domains, less surprisingly ancients assessed in plants and animal. Other claims are not convincing.

1) A main claim of the manuscript, that "events during the earliest stages of life continue to have an impact on the composition of ancient sequences" is not supported. The research, as described in the "general assessment" above, is focused on amino acid composition and other protein summary statistics compared among age of origin classes. Even if most of the analysis is accepted as correct, this data can only be interpreted as evidence for change if you assume the starting composition of the proteins in these age classes is the same.

2) No evidence was provided that differences in summary statistics were driven by selection, as opposed to slow relaxation to equilibrium. This should not be implied in the Discussion section. Further, there is no evidence that anything has reached any kind of "optimum". These unfounded claims, for me, detract from the overall analysis.

3) The difference in plant and animal recent disorder prediction measures is a good point to refute some previous analyses, but it contradicts the idea that disorder is necessarily a selection-drive property of new proteins.

4) The slope in excess clustering of hydrophobic amino acids is framed as being consistent, but this is an unconventional way of putting it given that the plant variation is so high, and the mean is in the opposite direction. Further, the ancient proteins are in the same direction, but barely so (just into apparent significance based on direction).

5) The effect size of the change in excess clustering among these groups is not big compared to the magnitude of the excess clustering overall. It was not clear to me that this is enough to matter, particularly over short time periods such as a few hundred million years. It would be good to demonstrate that this average magnitude effect would have a large impact on disorder or stickiness.

6) Because the clustering effect size is so small, I was not convinced that it would be difficult to alter the amino acid composition to this degree if required by selection. That is, my sense is that protein composition is moderately malleable to allow a steady state to be achieved fairly quickly (eg millions of years) such that the effect of origin age would not be detected on this time scale.

7) While the correlation in composition between transmembrane and non-transmembrane AA usage frequencies is impressive, three amino acids, G, A, and V, are strong outliers. These are the simplest three amino acids, and while they were possibly recruited early into the genetic code, the cited inference is fairly speculative, and was itself a prediction based largely on the amino acid simplicity. The "order of recruitment" correlation with slope (Figure 4) consists of 17 amino acids with little correlation, and these early three that have a large positive slope. This does not convince that age of recruitment continues to affect frequencies. There are many possible amino acid property vectors, and clarity on what has driven these differences is lacking.

---

## [Author Response]

[Editors’ note: The authors appealed the original decision. What follows is the authors’ response to the first round of review.]

Although the questions addressed in the manuscript are of potential interest to a broad community, the organization of the manuscript and some of the method descriptions make it difficult to know what are the specific questions being asked (Introduction)

We would like to appeal this decision. Reviewer 2 was very positive, while reviewer 1 misunderstood what the manuscript was about, as we detail in this document. We acknowledge weaknesses in our writing of the Introduction, which led reviewer 1 to the incorrect idea that our focus was on establishing the technical superiority of certain methodology, rather than broader biological findings that we study here using methodologies that have elsewhere been shown to be improvements. We have fixed this flaw in the Introduction. However, we believe the reviewer could have resolved their confusion about the central focus of the manuscript if they had referred back to the Title and Abstract, both of which we continue to find quite clear, and entirely incompatible with reviewer 1’s characterization of what the two questions of our paper are.

Guided by a focus on the wrong two questions, reviewer 1 provided detailed comments about peripheral findings (e.g. a great deal of focus on comparisons to previous whole-gene work in mouse found in Figure 1—figure supplement 2 and Figure 5—figure supplement 1) as though they were central, and made no specific comments or mentions of what were actually our central findings.

and how the data is compiled and analyzed (results). Some of the results appear to be significant but with very small effect sizes, which may require a more careful examination and interpretation.

We now more thoroughly discuss the magnitude of the effect sizes in the manuscript and their potential to be biologically relevant, drawing on the results of Kosinski et al., (2020) in order to convert effect sizes into fitness terms. We also note here that all the effects observed in this study are due to factors experienced during the de novo birth of protein sequences, rather than functional specificity or chance variation, and as such we do not expect large signal : noise ratios. We expect function to largely determine protein properties, a point we now make explicitly in the manuscript.

Since the study is focusing on protein domains, it would also be useful to define in the introduction what kind of protein domains are being analyzed. For many people, a protein domain has a tertiary structure, which means that by definition it is ordered. It is therefore not sure what a disorder predictor is predicting for such domains.

This was previously done in the second half of a paragraph in the Introduction, which gives our operational definition of domains as things present in the pfam database. Our use of pfams follows standard practice for high-throughput analysis of domains. We have split that paragraph into two in order to better highlight the definition and added the two sentences: “The pfam database has developed over time, and now curates sets of clearly homologous sequences that are no longer required to correspond to compact folds. We continue, however, to use the term “domain” in conjunction with pfams.”

Reviewer #1:Prior phylostratigraphic studies have reported correlations between various properties and gene age, where gene age is estimated from the last common ancestor of the species that harbor homologs of that gene. Failure to identify distant homologs is a source of error in gene age estimates. The prevalence of such errors, whether the error introduces systematic bias, and the impact of errors on phylostratigraphic inference is currently a subject of debate. This manuscript seeks to address these issues through the use of Pfam Hidden Markov models, instead of blastp searches, for homolog detection.

The switch from blastp to the hmms used by pfam is indeed one of the ways our manuscript is an advance on previous work. Another is the extensive quality filters that we introduce, e.g. to remove contaminants. A third, required for our major results, is the dramatic increase in the number of focal species relative to past phylostratigraphy studies.

The manuscript presents results that are relevant to two lines of inquiry:– Does the use of Pfam HMMs reduce error in gene age estimates via greater sensitivity?

This is not a question that our manuscript addresses, and as such was stated as fact, accompanied by citations from the Eddy lab. We take it as well-established prior knowledge that hmmer, when used with a multiple sequence alignment, has dramatically higher sensitivity than blastp performed in a pairwise manner, with no loss of specificity. We have now expanded upon this point in the introduction and have added additional references which compare homology detection methods, to emphasize the demonstrable improvement of HMM-based methods.

It is logically entailed that improving homology detection reduces errors in age estimates. We also cite the simulations of Moyers and Zhang as evidence that improvement in age estimates resulting from greater sensitivity will, as naively expected, reduce artefactual trends rather than strengthening them.

Upon careful re-reading of our submitted manuscript, we cannot understand how it could have been read as having this question as a central line of inquiry, but we nevertheless hope that these edits make this clearer.

Do the age estimates obtained with Pfam HMMs exhibit evidence of systematic bias that could lead to erroneous conclusions?

Something like this is indeed part of our study, albeit merely as a necessary validation step along the way to scientific insight rather than a primary scientific focus. However, we note that effect sizes matter here. Systematic bias much smaller in size than the actual trends would not be a problem, so our question is not, as phrased by reviewer 1, a question of the existence of bias. The question for our purposes is instead whether the trends we see are due to bias.

We have added a citation of related recent work from our lab (Kosinski et al., 2020) that validates our amino acid trends against an external experimental dataset on the fitness effects of peptides, further ruling out an important role for homology detection bias, and supporting our interpretation of phylostratigraphy trends as reflective of the process of gene birth.

– What do gene age estimates obtained using Pfam HMMs tell us about the forces that govern the evolution of amino acid composition in proteins of different ages?

This is not an accurate representation of the related question we address. We ask how protein sequences born at different times are different. Differences we find might be explained by the same forces governing their subsequent evolution but accumulating over different amounts of time, since they were all born with similar properties. Alternatively, they might be due to different conditions of life at the time of birth – as we conclude with respect to our oldest age groups.

We cannot (with the Materials and methods of this paper, this is something we are working on) and do not attempt to identify differences in subsequent evolutionary forces in proteins of different ages. We are uncomfortable in any case with the term “forces” in evolution, which in this case include regression to the mean – this could be seen as a force of mutation bias, but more causally important is the bias in the initial state. Reviewer 1 seems to have misunderstood the central questions and findings of the manuscript.

Both questions are important.

Beyond the details of these questions above, we agree that both the question of whether trends are real biological signals vs. artefacts, and the question of what evolutionary cause they have if real, are important. The reviewer has omitted any mention of the primary question we address (as clearly reflected in both our Title and our Abstract), which is whether patterns are the same vs. different in the 3 groups we address – animals, plants, and ancient protein sequences that predate the divergence of these two groups.

However, there are substantial difficulties with the analysis and interpretation, as presented here, that must be addressed before is possible to assess the quality of the evidence and how well it supports the conclusions with respect to either question.I) Lack of a formal hypothesis testing framework: In its current form, this manuscript does not address either of the above questions with sufficient formality and rigor to make a convincing case that the evidence supports a particular set of conclusions. To address this, the introduction should (1) summarize prior work, discuss open questions and unresolved controversies; (2) specifically state which hypotheses will be tested in the current study, with a discussion of the testable predictions that flow from these hypotheses; (3) describe how these predictions will be tested; and (4) what steps will be taken to eliminate confounding factors and rule out alternate hypotheses.

This is partly a matter of personal style. We have two main scientific questions rather than one question with a short enough list of hypotheses to be easily listed. Our two questions are not what the reviewer highlighted above, but are as follows: Question #1: what non-artefactual patterns are there in protein properties as a function of age? Question #2: are these patterns universal or taxon-specific? To use the framework preferred by this reviewer, and moreover to do so honestly, we would need to list as a separate hypothesis every conceivable pattern that our methods could uncover. This would not in our view improve clarity, given the large number of possible patterns considered in a combinatorial manner across our 3 taxonomic groups. There are just too many different possible outcomes to coherently list them all.

We agree that the Introduction needed a clearer focus on our two main questions. We have made substantial edits to fix this, in particular to what are now the fifth and last paragraphs, and believe it to now be much clearer.

For the first question (mitigation of error due to more accurate homology identification), steps 1 and 2 are handled reasonably well. However, testable predictions are mentioned haphazardly in the results and confounding factors are not adequately addressed. The second question (what forces act on the amino acid composition encoded by genes as they age) is not formally stated as a target of inquiry, but introduced almost in passing in the middle of the manuscript.

It would have been helpful for the reviewer to tell us where they thought it was introduced in passing in the middle of the manuscript, so we could clarify that this was not a question that the current manuscript is able to address. Phylostratigraphy trends are informative about the forces shaping gene birth, not about the forces changing genes as they age.

We made a number of changes to the Introduction to try to make this clearer earlier. The manuscript pivots at a certain point from confirming previously reported trends in ISD and clustering to discovering new trends in amino acid composition. This was previously introduced extremely briefly in the last paragraph of the Introduction. By splitting that paragraph in two, we have given this more emphasis, to better prepare the reader. We also added material to the fifth paragraph of the Introduction, to introduce broad hypotheses concerning patterns for the phylostratigraphic trends of the 20 amino acids.

II) Prior art: This manuscript states that "A key innovation is to use Pfam domains rather than BLASTp as our unit of homology". However, Pfam HMM models have been brought to bear on the question of homology detection and gene age estimates in a prior work that is not cited or discussed in the current manuscript: Jain et al., 2019. The authors will wish to read that article to determine to what extent it duplicates, contradicts or is complementary to the results in the current manuscript.

We now reference this paper in our manuscript. It primarily focuses on the question of orthology detection which, while also important, is not the main question in our manuscript, which deals with trends in structural (rather than functional, as in Jain et al.,) properties with age. In Jain et al., proteins are taken as the unit of homology, however the authors incorporate pfam information in order to parameterize their evolutionary models. This is considerably different to the way in which pfams were used in the present analysis as the unit of homology.

In addition, they may find that the Jain et al. article offers useful ideas for methodological refinements and/or casts new light on how the results of the current manuscript should be interpreted.

We note that the proposed traceability index in Jain et al. is based on the results of BlastP orthology searches, and so while relevant to older phylostratigraphy results it is not applicable to our manuscript, which use multiple alignment based HMM searches, which are able to more accurately detect even distant homology. Incorporating the Jain et al. methodology would be computationally expensive considering our much larger dataset, and is unlikely to result in an improvement in gene age estimation.

In particular, Jain et al., deal explicitly with lineage- and family-specific differences in evolutionary rates, topics that are not explored in depth in the current manuscript.

We do not address evolutionary rates in our manuscript because we do not study how proteins evolve over time, but only the factors influencing the birth process. See also our more detailed response below to a more detailed suggestion from reviewer 1 as to why we might want to look at evolutionary rates.

III) Methodology: The methodology in the current manuscript is not sufficiently well described to allow a reviewer to fully assess the results or another scientist to reproduce them. Technical terms are used without definition. Descriptions of how various quantities are actually calculated are lacking. Attention is given to statistical significance (p-values), but not to effect sizes.

We now define the abbreviation ISD upon first use – we were unable to determine which other technical terms the reviewer is referring to. We now also fully describe how phylostratigraphy slopes are calculated in both the Results and Materials and methods.

Effect sizes are represented in our paper by regression slopes. These were always visible in the Figures, and we have now increased the degree to which we discuss them in the text. Some of our arguments are based on the direct comparisons of effect sizes.

We consider slope (i.e. signal) to be the effect size rather than R^2^ (i.e. signal: noise ratio). We have also however also increased our discussion of R^2^ values. For these, visual representation takes the form of box plots.

The presentation relies heavily on summary statistics and derived data, in ways that may obscure trends in the underlying raw data.

We are not sure what is meant. The only processing we do of raw data prior to fitting the clearly described linear models is to take the average over all instances of a pfam such that a pfam constitutes only a single datapoint to the analysis. To treat them as separate datapoints would introduce a massive amount of pseudoreplication. To use a random effect term, as we did in Wilson et al., 2017 and Foy et al., 2019, is not computationally viable for the much larger number of species analyzed here.

Some confusion here may come from our box plots, which are shown only to enable the visualization of our raw data, and do not enter the statistical pipeline.

Our amino acid slopes do fall into the category of summary statistics / derived data. Using one summary statistic per amino acid is necessary in order to analyze their relationships with other sets of 20 numbers corresponding to other amino acid properties. We believe this illuminates rather than obscures. We do retain confidence intervals / weights for these downstream analyses of amino acid summary statistics.

Insufficient information is given about the data used, and it is not clear what data and supplementary information will be made available to the reader, other than the eight supplementary figures. Specific examples of these problems are given in the detailed comments below, but this is not an exhaustive list.

Extensive raw data tables were available at figshare, as reported in the manuscript.

IV) Demonstrating that the evidence supports the conclusions: The manuscript contains a number of strongly word assertions without sufficient demonstration that these conclusions are in fact supported by the evidence. The logic underlying the assertions is not spelled out and alternate explanations are not ruled out or even discussed. The manuscript tends to conflate the description of an observation with the interpretation of what that observation means.

We respond to more specific criticisms below, however we hope that our improved Introduction somewhat addresses these concerns, which we suspect stem from misunderstanding as to what we believe is known from prior literature vs. what questions the current study addresses.

– In one example of these issues, the manuscript concludes that the use of Pfam HMMs resulted in improved detection accuracy. While this is highly plausible, the evidence for this conclusion is not rigorously demonstrated.

This is explicitly assumed, not concluded, as the revised Introduction now makes clearer, citing published evidence in support.

The main evidence presented is that phylostratigraphy slopes for mouse genes are steeper when Pfam HMMs are used for homology detection, compared to blastp. However, the manuscript does not demonstrate that steeper slopes are incontrovertible evidence of more accurate homology detection.

These results in Figure 1—figure supplement 2 and Figure 5—figure supplement 1 are evidence that the trends are not artefactual. The logic is that when methods are improved, the effect size of the trend (i.e. the slope) gets larger.

A comparison of the gene sets obtained using the two methods would provide a more convincing and direct assessment. Is there a decrease in false negatives without an increase in false positives?

There is no ground truth available for the age of different genes. Without it, there is no assessing the absolute rates of false negatives and false positives. What the reviewer is asking for is logically impossible.

The properties of sequences in the two sets should also be examined for evidence of systematic bias.

Without there being a ground truth available for gene ages, we do not believe that comparisons of gene data sets could provide any evidence of bias.

We also note that the comparison of old vs. new is possible only for genes not pfams. This gene comparison was included in supplementary figures, rather than the main text, because it is not a major focus of our analysis, which is focused on pfams. It is simply one of multiple supporting pieces of evidence that the trends we look at are unlikely to be mere artifacts.

Our sets of genes and pfams are available in the Supplementary data on figshare, for anyone who wants to do their own analysis.

The discussion of this issue exemplifies the tendency to confuse cause and effect. In the Results section, the statement "Our improved methodology increased the steepness of the relationship between gene ISD and gene age…" does not make a clear distinction between the observation (steeper slopes) and the inference (that steeper slopes are due to the new methodology and that the new methodology is an improvement). The inference must be demonstrated, not simply stated.

As discussed above, the improved nature of the methodology is not an inference but a premise. That steeper slopes are obtained with the new methodology is an observation, which supports the inference that the slopes are “real” i.e. have a biological and not artefactual cause.

– In a second example, the argument is made that the observed relationships between amino acid composition and age are unaffected by systematic error in homology detection because different trends are observed in plant and animal protein domains and "homology detection bias is expected to create similar patterns for all taxa". No argument is presented to support the statement that "homology detection bias is expected to create similar patterns for all taxa." While this might hold under some conditions, it is not clear to me why this should be true in taxonomic lineages that differ substantially in GC content or evolutionary rates.

Homology detection bias is defined as what happens when some types of sequences are systematically more prone to undetectable homology, leading to systematic error in which sequences are assigned too young an age. It has the potential to affect our trends when the protein property of interest is correlated with the ability to detect homology.

If one taxonomic group has a faster evolutionary rate, then homology detection becomes harder, and so there is a greater opportunity for homology detection bias within that group. But this does not change whether or not the property of interest is substantially correlated with the ability to detect homology.

Similarly, a %GC that has extreme divergence away from 50% might make homology detection harder. This seems in any case moot, because the taxonomic lineages in question are “plants”, “animals” and “pre-LECA”, which do not systematically differ from one another in this way. We have now added a section to the Discussion to address this point.

We do discuss at length the possibility that more changeable amino acids make it harder to detect homology, which has more potential to be a confounding factor in our work. We note that it is standard practice to use the same BLOSUM substitution matrix, regardless of which taxonomic group is being studied. This shows that the consensus in the field is that the factors that would change amino acid compositions which are most prone to homology detection bias do not vary substantially among taxa.

Even assuming that the "similar patterns for all taxa" prediction is valid, only one taxonomic comparison, plants versus animals, is offered as evidence.

Our third group is inferred pre-LECA ancestors, so in fact there are 3 pairwise comparisons across 3 taxonomic groups, with none of them showing similarity in amino acid trends.

If subsets of animal species are compared, for example, are dissimilar patterns also observed? Further, could the different patterns in plants and animals be due to issues with the underlying data? The Materials and methods section describes a substantial effort to obtain high quality genomes. (This is one of the strongest sections in the Materials and methods.) Despite, and perhaps because of, this effort, the number of animal genomes exceeds the number of plant genomes by almost a factor of four. Domain discovery, modeling, and annotation in plant genomes lags substantially behind animal genomes, in part because of a the relative dearth of proteomic data in plants. The fact that the slope for recent plant domains is positive and not statistically significant (p=0.1), as well as the large variance in plant slopes shown in Figure 1B, all suggest that there may be problems with the plant data that could be responsible for different trends.

We agree with the reviewer, and our manuscript already stressed, that the plant analyses (including but not limited to the ISD slope mentioned above) are underpowered. The same would be true if e.g. we were to compare vertebrates to insects – there is too little data on the latter. We note that the ancient trends, which the reviewer does not comment on, do have sufficient power, and that they can be compared to the animal trends. Note that Figure 2A shows that the ancient trends can be pulled out just as easily from the plant data as from the animal data, suggesting that the problem is with the number of plant-specific protein domains, not the plant genomes per se.

The manuscript contains other, similar problems with interpretation of evidence, in addition to the two examples given above.

While a more detailed list would of course be welcome, we think our responses to the examples above illustrate that the problem seems on a systematic basis to be one of style and communication, in which the purpose of the study was not clear to the reviewer, rather than our errors in the interpretation of evidence (for a purpose that the reviewer thinks we had or should have had, but which we did not). Our edits to the Introduction should have fixed the communication issue, clarifying the purpose of our study.

Reviewer #2:James et al., comprehensively outline the challenges in phylostratigraphy and homology detection while carefully applying these methods to detect age-dependent trends in protein sequences. Their method improves signal for previously reported trends such as decreased hydrophobicity and increased hydrophobic clustering in young protein sequences. Interestingly, the authors find increased intrinsic structural disorder (ISD) in young animal domains, but not young plant domains. Their method also allows them to gain insights into changes in amino acid frequency with gene age. These results would be of general interest to eLife readers. The following could help clarify the scope of these results:1) The authors should elaborate on the diversity of the taxa that are included. Although there are an impressive 435 species included in the analysis, there are only 5 non-plant and non-animal species, all of which are fungi. This is understandably due to quality control for data that are included in the analysis, but it's not clear if it's fair to use the term "universal" for the trends observed throughout the paper if they are based (mostly) on plants and animals.

We have added a supplementary figure outlining the phylogeny of species used in this analysis. We note that the ancient pfam domains included in our analysis predate the split of animals/fungi and plants, with some estimated to have been present in LUCA. Our claim to universality is based on being present in 3 lineages: animal, plant, and pre-LECA. We have edited the Abstract and other places to make clearer the fact that all reported trends, including those that predate eukaryotes, are assessed in eukaryotes.

2) The authors should comment on whether or not there are systematic differences in the plant vs. animal species that are included, and how that could affect the results of the study. For example, is the GC content of the included plant genomes a concern? There seem to be some reported differences in GC content of different plant species (monocots vs. dicots) [Kawabe and Miyashita, 2003; Li and Du, 2014, Šmarda et al., 2014] - are these species broadly sampled in this study? If not, this should be clearly stated.

We now discuss possible differences in genomic GC content between the two groups. Thank you for the references, which we include.

[Editors’ note: what follows is the authors’ response to the second round of review.]

Reviewer #3:Trends in summary statistics of proteins between plants and animals based on grouping age of origin show that bias in detecting homologous proteins is accounted for and not the strongest effect in results. Authors convincingly show (1) there are minor differences in degree of clustering of hydrophobic residues between proteins with ancient origins and more recent animal proteins; (2) animal and plant proteins with more recent origins have different degrees of predicted disorder; (3) Amino acid frequency trends are surprisingly consistent between ancient transmembrane and non-transmembrane domains, less surprisingly ancients assessed in plants and animal. Other claims are not convincing.

We thank the reviewer for their comments and we believe that we have addressed their concerns. Our interpretations of results have been softened throughout the revised manuscript, with particular attention to what the role of selection might or might not have been. Detailed responses are given below.

1) A main claim of the manuscript, that "events during the earliest stages of life continue to have an impact on the composition of ancient sequences" is not supported. The research, as described in the "general assessment" above, is focused on amino acid composition and other protein summary statistics compared among age of origin classes. Even if most of the analysis is accepted as correct, this data can only be interpreted as evidence for change if you assume the starting composition of the proteins in these age classes is the same.

The authors seem to agree that the results "are not evidence of composition change" but "due to sequences born at different times". In the early version I looked at, I interpreted the main claim about "impact from early events" to be arguing for composition change. I think we are in agreement about this now.

We agree with the reviewer that our results are not evidence for composition change; our interpretation (“main claim” above) is that proteins born at different times have different amino acid compositions, due to the differing availability of amino acids at different stages of life. We have made this interpretation clearer in the revised manuscript, both in the abstract and in the Results section.

2) No evidence was provided that differences in summary statistics were driven by selection, as opposed to slow relaxation to equilibrium. This should not be implied in the Discussion section. Further, there is no evidence that anything has reached any kind of "optimum". These unfounded claims, for me, detract from the overall analysis.

The Discussion still states that “Our results are consistent with the hypothesis that selection acts to reduce the aggregation propensity of proteins.” This weak claim remains true, and the remainder of this paragraph cites other work (not the current manuscript) in support. We have been more careful in our discussion of selection, removed mentions of “optima”, and added explicit discussion of the possibility of relaxation to (mutational) equilibrium.

3) The difference in plant and animal recent disorder prediction measures is a good point to refute some previous analyses, but it contradicts the idea that disorder is necessarily a selection-drive property of new proteins.

We do not believe that the non-universality of trends is evidence against selection. Neither do we believe that if we had found trends to be universal, that would on its own have been evidence for selection. We see universality and selection as separate questions. We discuss the possibility that de novo proteins in plants may be under different selection pressures, due to the presence of aggregation inhibiting molecules such as polyphenols in plant cells, and we now include a mention of this in the Results section so as to remove any apparent contradiction.

4) The slope in excess clustering of hydrophobic amino acids is framed as being consistent, but this is an unconventional way of putting it given that the plant variation is so high, and the mean is in the opposite direction. Further, the ancient proteins are in the same direction, but barely so (just into apparent significance based on direction).

This section has been altered to be more intuitive and stressing that we have little power in plant data. However, it is clear that the overall trend in clustering overlaps the trend for ancient domains, which is not the case for the trends in ISD. While the error bars are larger, the trend remains highly significant for ancient domains (see Figure 5—figure supplement 2), not barely so as the reviewer states. We have included in the Results text the slopes and p-values for each of the two significant taxonomic groups in order to assure the reader of the significance of this result, in addition to continuing to display slopes and associated confidence intervals in Figure 5B. We also added values for the number of data points (n) to the Figure 5 legend.

5) The effect size of the change in excess clustering among these groups is not big compared to the magnitude of the excess clustering overall. It was not clear to me that this is enough to matter, particularly over short time periods such as a few hundred million years.

In the previously revised version of the manuscript that the reviewer did not see we highlighted and interpreted the small effect size of our result in the second paragraph of the Discussion.

It would be good to demonstrate that this average magnitude effect would have a large impact on disorder or stickiness.

Metrics of disorder or stickiness are dominated by amino acid composition, which clustering excludes by design. We have added confirmation of the lack of a significant correlation between clustering and disorder. Our interest in clustering come precisely because it may affect a variety of protein properties in a manner that is independent from amino acid frequencies. We have improved our discussion of the clustering metric in the Results section to avoid confusion.

6) Because the clustering effect size is so small, I was not convinced that it would be difficult to alter the amino acid composition to this degree if required by selection.

Similarly, in point #6, the confusion may have to do with communication about ongoing compositional change. In addressing this point though, it would be good to clarify what is meant by "composition". I understood it to be quite general, such that composition change would include the substitutions required to affect clustering. When the authors say clustering is "not affected by amino acid composition", I think they are using it to mean something narrower, equivalent to amino acid frequencies. I don't think we are disagreeing here, but miscommunicating, possibly due to my lack of precision in word choice for which I apologize. I want to see some citation or evidence, though, if the authors are going to argue that it requires "many", but I think we're roughly in agreement here. The authors' point out that there are differences in the proteins that lead to this clustering difference (no matter the number of amino acid substitution required) stands.

We agree this may be a miscommunication. Yes, by “amino acid composition” we mean amino acid frequencies, and we have replaced this term throughout the manuscript. The distinction between amino acid frequencies and order is now clearer. Our clustering metric is designed to be independent from the former, in order to allow focused study of the latter.

Changing clustering requires many substitutions distributed across the entire protein-coding sequence. We have included further discussion of the clustering metric in the revised manuscript. Largely, we refer the reader/reviewer to Bertram and Masel, 2020, a manuscript devoted in large part to this point.

That is, my sense is that protein composition is moderately malleable to allow a steady state to be achieved fairly quickly (eg millions of years) such that the effect of origin age would not be detected on this time scale.

As we do detect systematic trends in both clustering and amino acid composition with protein age, even among ancient domains, we are unsure how to interpret the reviewer’s comment. This is a major reason why we think our findings are so interesting – because they do not confirm the intuition of the reviewer and others. Empirically, we found that clustering has not reached a steady state. The fact that our finding is so unintuitive in being found at all is what makes it remarkable, despite its small effect size. We discuss several interpretative hypotheses as to what could account for this result.

Even if there were a steady state, it would not be one by mutation but by mutation-selection balance: hydrophobic amino acids are more evenly dispersed than would be expected from chance in older proteins. We now fully discuss this point in the seventh paragraph of the Discussion.

7) While the correlation in composition between transmembrane and non-transmembrane AA usage frequencies is impressive, three amino acids, G, A, and V, are strong outliers.

We assume the reviewer is referring to these as outliers in terms of the magnitude of trends in frequencies. In the Figure 3 analysis, we use Spearman’s ranked correlation coefficient, which disregards these magnitudes. In addition, we note these amino acids fall very neatly along the dashed x=y line shown in Figure 3A.

These are the simplest three amino acids, and while they were possibly recruited early into the genetic code, the cited inference is fairly speculative, and was itself a prediction based largely on the amino acid simplicity.

The inferred order of recruitment is indeed speculative by its nature, which we have highlighted in the revised manuscript (Results,). However, simplicity does seem a reasonable predictor of order of recruitment into biological use.

The "order of recruitment" correlation with slope (Figure 4) consists of 17 amino acids with little correlation, and these early three that have a large positive slope. This does not convince that age of recruitment continues to affect frequencies. There are many possible amino acid property vectors, and clarity on what has driven these differences is lacking.

On point 7, I am less certain but think we are in agreement. I did not mean to imply the effect should be thrown out, but I think the authors get the point that with three amino acids (GAV) almost wholly responsible for the effect, inferences about which physicochemical properties are responsible for driving the effect are highly speculative, even if reasonable. It sounds like the authors intend to highlight the possible alternatives, so I think this aspect is likely to be fine.

In the analyses shown in Figure 4 we again use Spearman’s, and so the magnitude of the slope will not affect the analysis. It is not surprising that a Spearman’s correlation among 20 points can be destroyed by selectively removing the 3 points that *a posteriori* most contribute to the correlation, especially when those are the 3 points with the strongest signal in a parametric analysis. This fact is captured within the Figure 4a Spearman’s rho = -0.58, p = 0.008.